# COLLABORATION! TOWARDS ROBUST NEURAL METHODS FOR ROUTING PROBLEMS

## ABSTRACT

While enjoying desirable efficiency and less dependence on domain expertise, existing neural methods for vehicle routing problems (VRPs) are vulnerable to adversarial attacks – their performance drops drastically on adversarial instances, i.e., clean instances with crafted perturbations. To enhance the robustness, we propose a *Collaborative Neural Framework (CNF)* w.r.t the adversarial defense of neural methods for VRPs, which is crucial yet underexplored in literature. Given a neural method, we adversarially train multiple models in a collaborative manner to synergistically promote the robustness against attacks, while maintaining (or even boosting) the standard generalization on clean instances. A neural router is designed to elegantly distribute instances to each model, which improves overall load balancing and collaborative performance. Extensive experiments verify the effectiveness and versatility of CNF to defend against various attacks for different neural methods. Notably, our trained models also achieve decent out-of-distribution generalization performance on real-world benchmark instances.

## 1 INTRODUCTION

Combinatorial optimization problems (COPs) are crucial yet challenging to solve due to the NP-hardness. Neural combinatorial optimization (NCO) aims to leverage machine learning (ML) to explore powerful heuristics for solving COPs, and has attracted considerable attention recently (Bengio et al., 2021). Among them, a large number of NCO works develop neural methods for *vehicle routing problems* (VRPs) – one of the most classic COPs with broad applications in transportation (Pillac et al., 2013), logistics (Konstantakopoulos et al., 2022), planning and scheduling (Padrón et al., 2016), etc. With supervised or reinforcement learning (RL), the neural methods could learn construction or improvement heuristics for VRPs, which achieve competitive or even superior performance to the conventional algorithms. However, recent studies show that these neural methods suffer from severe adversarial robustness issue (Geisler et al., 2022), where their performance drops drastically on clean instances (sampled from the training distribution) with crafted perturbations.

Although the adversarial robustness has been investigated in a couple of recent works (Zhang et al., 2022; Geisler et al., 2022; Lu et al., 2023), the defensive methods on how to help forge sufficiently robust neural methods for VRPs are still underexplored. In particular, existing endeavours mainly focus on the *attack* side, where they propose different perturbation models to generate adversarial instances. On the *defense* side, they simply follow the vanilla adversarial training (AT) (Madry et al., 2018). Concretely, treated as a *min-max* optimization problem, they first generate adversarial instances that maximally degrade the current model performance, and then minimize the empirical losses of these adversarial variants. However, we empirically observe that they may suffer from the undesirable trade-off (Tsipras et al., 2019; Zhang et al., 2019) between standard generalization (on clean instances) and adversarial robustness (against adversarial instances) by simply adhering to the vanilla AT. As demonstrated in Fig. 1, the vanilla AT improves adversarial robustness of the neural solver at the expense of standard generalization (e.g., POMO (1) vs. POMO_AT (1)). We also identify a similar issue from the empirical results of Lu et al. (2023). One of the major reasons is that the training model is not sufficiently expressive (Geisler et al., 2022). We empirically justify this viewpoint by simply increasing the model capacity (i.e., adversarially training multiple models), which can partially mitigate the trade-off (e.g., POMO_AT (1) vs. POMO_AT (3)). However, it is still an open challenge on *how to effectively synergize multiple models to achieve favorable overall performance on both clean and adversarial instances within a reasonable computational budget.*

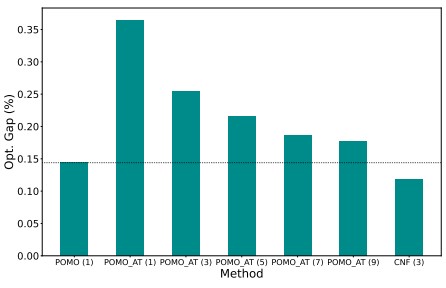

(a) Performance on Clean Instances

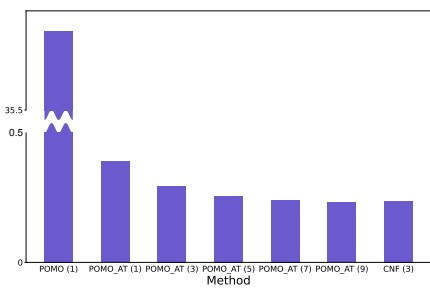

(b) Performance on Adversarial Instances

Figure 1: **Performance of POMO (with vanilla AT) on TSP100 with the attack method in Zhang et al. (2022).** The value in brackets denotes the number of trained models. During inference, we choose the best result (among all trained models) for each instance, and report the average optimality (opt.) gap over 1000 instances. The results reveal the vulnerability of existing neural methods to adversarial attacks, and the existence of the trade-off between standard generalization and adversarial robustness in VRPs. Similar phenomenon could also be observed on CVRP as shown in Table 1. Details of the attack method and setups can be found in Appendix B.1 and Section 4, respectively.

In this paper, we focus on the *defense* of neural methods for VRPs, with the aim to concurrently enhance both the standard generalization and adversarial robustness. Instead of separately training multiple models, we propose a *Collaborative Neural Framework (CNF)* to exert the AT on multiple models in a collaborative manner. Specifically, in the inner maximization optimization of CNF, we synergize multiple models to further generate the *global* adversarial instance for each clean instance by attacking the best-performing model, rather than only leveraging each model to independently generate their own *local* adversarial instances. In doing so, the generated adversarial instances are diverse and strong in benefiting the policy exploration and attacking the models, respectively (see Section 3). In the outer minimization optimization of CNF, we train an attention-based neural router to forward instances to models for training, which helps achieve satisfactory load balancing and collaborative performance. The overview of CNF is illustrated in Fig. 2.

Our contributions are outlined as follows. 1) In contrast to the recent endeavors on the attack side, we focus on the defense of neural methods for VRPs, which is crucial yet underexplored in literature. We empirically observe that the defense through the vanilla AT may lead to the undesirable trade-off between standard generalization and adversarial robustness in VRPs, even if simply increasing the model capacity (see Fig. 1). 2) We propose a collaborative neural framework (CNF) to concurrently enhance the performance on both clean and adversarial instances. Specifically, we propose to further generate global adversarial instances, and design a neural router to distribute instances to each model for effective training. 3) We evaluate the effectiveness and generalizability of our method against various attacks on different VRPs, such as the (symmetric and asymmetric) traveling salesman problem (TSP) and capacitated vehicle routing problem (CVRP). Results show that our framework could greatly improve the adversarial robustness of neural methods while preserving (or even improving) the standard generalization. Beyond the expectation, we also observe the boosted out-of-distribution (OOD) generalization on both synthetic and benchmark instances, which may suggest the favorable potential of our method in promoting various types of generalization of neural VRP methods.

## 2 PRELIMINARIES

We first present the definition of VRPs and the neural methods to learn autoregressive construction heuristics (Kool et al., 2018; Kwon et al., 2020), then we introduce the standard AT (Madry et al., 2018) and its challenges for the discrete VRPs.

### 2.1 NEURAL METHODS FOR VRPS

**Problem Definition.** Without loss of generality, we define a VRP instance $x$ over a graph $\mathcal{G} = \{\mathcal{V}, \mathcal{E}\}$, where $\mathcal{V} = \{v_i\}_{i=1}^n$ represents the node set, and $(v_i, v_j) \in \mathcal{E}$ represents the edge set with $v_i \neq v_j$. The solution $\tau$ to a VRP instance is a tour, i.e., a sequence of nodes in $\mathcal{V}$. The cost function

$c(\cdot)$ computes the total Euclidean length of a given tour. The objective is to seek an optimal tour $\tau^*$ with the minimal cost: $\tau^* = \arg\min_{\tau \in \Phi} c(\tau|x)$, where $\Phi$ is the set of all feasible tours which obey the problem-specific constraints. For example, a feasible tour in TSP should visit each node exactly once, and return to the starting node in the end. For CVRP, each node in $\mathcal{V}$ is associated with a demand $\delta_i$, and a depot node $v_0$ is additionally set with zero demand. Given the capacity $Q$ for each vehicle, a tour in CVRP consists of multiple sub-tours, each of which represents a vehicle starting from the depot, visiting a subset of nodes in $\mathcal{V}$ and returning to the depot. It is feasible if each node in $\mathcal{V}$ is visited exactly once, and the total demand in each sub-tour is upper bounded by $Q$.

**Neural Autoregressive Methods.** Popular neural methods (Kool et al., 2018; Kwon et al., 2020) construct a solution to a VRP instance following Markov Decision Process (MDP), where the policy is parameterized by a neural network with parameters $\theta$. The policy takes the states as inputs, which are instantiated by features of the instance and the partially constructed solution. Then, it outputs the probability distribution of valid nodes to be visited next, from which an action is taken by either greedy rollout or sampling. After a complete tour $\tau$ is constructed, the probability of the tour can be factorized via the chain rule as $p_\theta(\tau|x) = \prod_{s=1}^{S} p_\theta(\pi_\theta^s | \pi_\theta^{<s}, x)$, where $\pi_\theta^s$ and $\pi_\theta^{<s}$ represent the selected node and the partial solution at the $s_{\text{th}}$ step, and $S$ is the number of total steps. Typically, the reward is defined as the negative length of a tour $\mathcal{R} = -c(\tau|x)$. The policy network is commonly trained with REINFORCE (Williams, 1992) algorithm. With a baseline function $b(\cdot)$ to reduce the gradient variance and stabilize the training, it estimates the gradient of the expected reward $\mathcal{L}(\theta|x) = \mathbb{E}_{p_\theta(\tau|x)} c(\tau)$ such that:

$$\nabla_\theta \mathcal{L}(\theta|x) = \mathbb{E}_{p_\theta(\tau|x)}[(c(\tau) - b(x))\nabla_\theta \log p_\theta(\tau|x)]. \tag{1}$$

## 2.2 ADVERSARIAL TRAINING

Adversarial training is one of the most effective and practical techniques to equip deep learning models with adversarial robustness against crafted perturbations on the clean instance. In the supervised fashion, where the clean instance $x$ and ground truth (GT) label $y$ are given, AT is commonly formulated as a min-max optimization problem:

$$\min_\theta \mathbb{E}_{(x,y) \sim \mathcal{D}}[\ell(y, f_\theta(\tilde{x}))], \text{ with } \tilde{x} = \arg\max_{\tilde{x}_i \in \mathcal{N}_\epsilon[x]}[\ell(y, f_\theta(\tilde{x}_i))], \tag{2}$$

where $\mathcal{D}$ is the data distribution; $\ell$ is the loss function; $f_\theta(\cdot)$ is the model prediction with parameters $\theta$; $\mathcal{N}_\epsilon[x]$ denotes the neighborhood around $x$, with its size constrained by the attack budget $\epsilon$. The solution to the inner maximization is typically approximated by projected gradient descent (PGD):

$$x^{(t+1)} = \Pi_{\mathcal{N}_\epsilon[x]}[x^{(t)} + \alpha \cdot \texttt{sign}(\nabla_{x^{(t)}} \ell(y, f_\theta(x^{(t)})))], \tag{3}$$

where $\alpha$ is the step size; $\Pi$ is the projection operator that projects the adversarial instance back to the neighborhood $\mathcal{N}_\epsilon[x]$; $x^{(t)}$ is the adversarial instance found at step $t$; and the $\texttt{sign}$ operator is used to take the gradient direction and carefully control the attack budget. Typically, $x^{(0)}$ is initialized by the clean instance or randomly perturbed instance with small Gaussian or Uniform noises. The adversarial instance is updated iteratively towards loss maximization until a stop criterion is satisfied.

**AT for VRPs.** Most ML research on adversarial robustness focuses on the continuous image domain (Goodfellow et al., 2015; Madry et al., 2018). We would like to highlight two main differences in the context of discrete VRPs (or COPs). 1) *Imperceptible perturbation:* The adversarial instance $\tilde{x}$ is typically generated within a small neighborhood of the clean instance $x$, so that the adversarial perturbation is imperceptible to human eyes. For example, the adversarial instance in image related tasks is typically bounded by $\mathcal{N}_\epsilon[x] : \|x - \tilde{x}\|_p \leq \epsilon$ under the $l_p$ norm threat model. When the attack budget $\epsilon$ is small enough, $\tilde{x}$ retains the GT label of $x$. However, it is not the case for VRPs due to the nature of discreteness. The optimal solution could be significantly changed even if only a small part of the instance is modified. Therefore, the subjective imperceptible perturbation is not a realistic goal in VRPs, and we do not exert such an explicit imperceptible constraint on the perturbation model. 2) *Accuracy-robustness trade-off*: The standard generalization and adversarial robustness seem to be conflicting goals in image related tasks. With increasing adversarial robustness the standard generalization tends to decrease, and a number of works intend to mitigate such accuracy-robustness trade-off in this domain (Tsipras et al., 2019; Zhang et al., 2019; Wang et al., 2020; Raghunathan et al., 2020; Yang et al., 2020b). By contrast, a recent work (Geisler et al., 2022) claims the existence of neural solvers with high accuracy and robustness in COPs. They demonstrate that a *sufficiently*

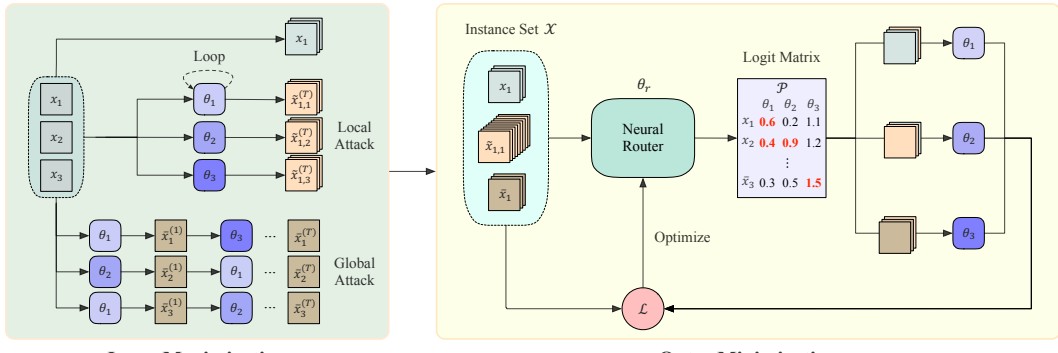

Figure 2: The overview of the proposed framework. Suppose we train $M = 3$ models on a batch ($B = 3$) of clean instances. The inner maximization generates local ($\tilde{x}$) and global ($\bar{x}$) adversarial instances within $T$ steps. In the outer minimization, a neural router $\theta_r$ is jointly trained to distribute instances to the $M$ models for training. Specifically, based on the logit matrix $\mathcal{P}$ predicted by the neural router, each model selects the instances with Top$\mathcal{K}$-largest logits (e.g., red ones). The neural router is optimized to maximize the improvement of collaborative performance after each training step of $\Theta$. For simplicity, we omit the superscripts of instances in the outer minimization.

*expressive* model does not suffer from the trade-off given the problem-specific *efficient and sound* perturbation model, which guarantees the correct GT label of the perturbed instance. Empirically, by following the vanilla AT, we observe the undesirable trade-off may still exist (as shown in Fig. 1), which is mainly due to the insufficient model capacity under the specific perturbation model.

## 3 COLLABORATIVE NEURAL FRAMEWORK

In this section, we first present the motivation and overview of the proposed method, and then introduce the technical details. Overall, we propose a collaborative neural framework to synergistically promote adversarial robustness among multiple models, while maintaining decent standard generalization. Since conducting AT for deep learning models from scratch is computationally expensive due to the extra inner maximization steps, we use the model pretrained on clean instances as a warm-start for subsequent AT steps. The overview of the proposed method is illustrated in Fig. 2.

**Motivation.** Motivated by the empirical observations that 1) existing neural methods for VRPs suffer from the severe adversarial robustness issue; 2) undesirable trade-off between adversarial robustness and standard generalization may exist when following the vanilla AT (even if increasing the model capacity), we propose to adversarially train multiple models in a *collaborative* manner to mitigate the above-mentioned issues within a reasonable computational budget. It then raises the research question on how to effectively and efficiently train multiple models under the AT framework, involving a pair of inner maximization and outer minimization, which will be detailed in the following parts. Note that despite the accuracy-robustness trade-off is a well-known research problem in the literature of adversarial ML, most works focus on the continuous image domain. Due to the needs for GT labels or the dependence on the imperceptible perturbation model, their methods (e.g., TRADES (Zhang et al., 2019), Robust Self-Training with AT (Raghunathan et al., 2020)) cannot be directly adapted to the discrete VRP domain. We refer to Appendix E for further discussions.

**Overview.** Given the pretrained model $\theta_p$, CNF deploys the AT on its $M$ copies (i.e., $\Theta^{(0)} = \{\theta_j^{(0)}\}_{j=1}^M \leftarrow \theta_p$) in a collaborative manner. At each training step, it first solves the inner maximization optimization to synergistically generate the local ($\tilde{x}$) and global ($\bar{x}$) adversarial instances, on which the current models underperform. Then, in the outer minimization optimization, we jointly train a neural router $\theta_r$ with all models $\Theta$ by RL in an end-to-end manner. By adaptively distributing all instances to different models for training, the neural router could reasonably exploit the overall capacity of models and thus achieve satisfactory load balancing and collaborative performance. During inference, we discard the neural router $\theta_r$ and use the trained models $\Theta$ to solve each instance. The best solution among them is returned to reflect the final *collaborative performance*. We present the pseudocode of CNF in Algorithm 1, and elaborate each part in the following subsections.

---

**Algorithm 1** Collaborative Neural Framework for VRPs

---

**Input:** training steps: $E$, number of models: $M$, attack steps: $T$, batch size: $B$, pretrained model: $\theta_p$;
**Output:** robust model set $\Theta^{(E)} = \{\theta_j^{(E)}\}_{j=1}^{M}$;

1: Initialize $\Theta^{(0)} = \{\theta_1^{(0)}, \cdots, \theta_M^{(0)}\} \leftarrow \theta_p$
2: **for** e = 1, ..., $E$ **do**
3:      $\{x_i\}_{i=1}^{B} \leftarrow$ Sample a batch of clean instances
4:      Initialize $\tilde{x}_{i,\cdot}^{(0)}, \bar{x}_i^{(0)} \leftarrow x_i, \quad \forall i \in [1, B]$
5:      **for** t = 1, ..., $T$ **do**                                ▷ *Inner Maximization*
6:          $\tilde{x}_{i,j}^{(t)} \leftarrow$ Approximate the solutions of $\max \ell(\tilde{x}_{i,j}^{(t-1)}; \theta_j^{(e-1)}), \quad \forall i \in [1, B], \forall j \in [1, M]$
7:          $\theta_{b(i)}^{(e-1)} \leftarrow$ Choose the best-performing model for $\bar{x}_i^{(t-1)}$ from $\Theta^{(e-1)}, \quad \forall i \in [1, B]$
8:          $\bar{x}_i^{(t)} \leftarrow$ Approximate the solutions of $\max \ell(\bar{x}_i^{(t-1)}; \theta_{b(i)}^{(e-1)}), \quad \forall i \in [1, B]$
9:      **end for**
10:      $\mathcal{X} \leftarrow \left\{ \{x_i, \tilde{x}_{i,j}^{(T)}, \bar{x}_i^{(T)}\}, \quad \forall i \in [1, B], \forall j \in [1, M] \right\}$          ▷ *Outer Minimization*
11:      $\mathcal{R} \leftarrow$ Evaluate $\mathcal{X}$ on $\Theta^{(e-1)}$
12:      $\tilde{\mathcal{P}} \leftarrow \texttt{Softmax}(f_{\theta_r^{(e-1)}}(\mathcal{X}, \mathcal{R}))$
13:      $\Theta^{(e)} \leftarrow$ Train $\Theta^{(e-1)}$ on $\texttt{Top}\mathcal{K}(\tilde{\mathcal{P}})$ instances
14:      $\mathcal{R}' \leftarrow$ Evaluate $\mathcal{X}$ on $\Theta^{(e)}$
15:      $\theta_r^{(e)} \leftarrow$ Update neural router $\theta_r^{(e-1)}$ with the gradient $\nabla_{\theta_r^{(e-1)}} \mathcal{L}$ using Eq. (6)
16: **end for**

---

## 3.1 INNER MAXIMIZATION

The inner maximization aims to generate adversarial instances for training in the outer minimization, which should be 1) effective in attacking the models $\Theta$; 2) diverse to benefit the policy exploration for VRPs. Typically, an iterative attack method generates local adversarial instances for each model only based on its own parameter (e.g., the same $\theta$ in Eq. (3) is repetitively used throughout the generation). Such *local attack* (line 6) only focuses on degrading each individual model, failing to consider the ensemble effect of multiple models. Due to the existence of multiple models in CNF, we are motivated to further develop the *global attack* (line 7-8), where each adversarial instance could be generated using different model parameters. Concretely, given each input (clean) instance $x$, we generate the global adversarial instance $\bar{x}$ by attacking the corresponding *best-performing model* in each iteration of the inner maximization. In doing so, compared with the sole local attack, the generated adversarial instances are more diverse to successfully attack the models $\Theta$ (see Appendix E for further discussions). Without loss of generality, we take the attacker from Zhang et al. (2022) as an example, which directly maximizes the variant of the reinforcement loss as follows:

$$\ell(x; \theta) = \frac{c(\tau)}{b(x)} \log p_\theta(\tau | x), \tag{4}$$

where $b(\cdot)$ is the baseline function (as shown in Eq. (1)). On top of it, we generate the global adversarial instance $\bar{x}$ such that:

$$\bar{x}^{(t+1)} = \Pi_{\mathcal{N}}[\bar{x}^{(t)} + \alpha \cdot \nabla_{\bar{x}^{(t)}} \ell(\bar{x}^{(t)}; \theta_b^{(t)})], \quad \theta_b^{(t)} = \arg\min_{\theta \in \Theta} c(\tau | \bar{x}^{(t)}; \theta), \tag{5}$$

where $\bar{x}^{(t)}$ is the global adversarial instance and $\theta_b^{(t)}$ is the best-performing model (w.r.t. $\bar{x}^{(t)}$) at step $t$. If $\bar{x}^{(t)}$ is out of the range, it would be projected back to the valid domain $\mathcal{N}$ by $\Pi$, such as the min-max normalization for continuous variables (e.g., node coordinates) or another rounding operation for discrete variables (e.g., node demands). Here we discard the `sign` operator in Eq. (3) to relax the imperceptible constraint in VRPs. More details are presented in Appendix B.1.

In summary, the local attack is a special case of the global attack where the same model is chosen as $\theta_b$ in each iteration. While the local attack aims to degrade a single model, the global attack could be viewed as explicitly attacking the collaborative performance of multiple models $\Theta$, which takes into consideration the ensemble effect by attacking $\theta_b$. In CNF, we involve adversarial instances that are generated by both the local and global attacks to pursue better adversarial robustness. We also collect clean instances as done in Zhang et al. (2022) to preserve the standard generalization.

### 3.2 OUTER MINIMIZATION

After the adversarial instances are generated by the inner maximization, we collect a set of instances $\mathcal{X}$ with $|\mathcal{X}| = N$, which includes clean instances $x$, local adversarial instances $\tilde{x}$ and global adversarial instances $\bar{x}$, to train $M$ models. Here a key problem is that *how are the instances distributed to models for their training, so as to achieve satisfactory load balancing (training efficiency) and collaborative performance (effectiveness)?* To solve this, we design an attention-based neural router, and jointly train it with the models $\Theta$ to maximize the improvement of collaborative performance.

Concretely, we first evaluate the $N$ instances on each model to obtain a cost matrix $\mathcal{R} \in \mathbb{R}^{N \times M}$. The neural router $\theta_r$ takes as inputs the instances $\mathcal{X}$ and $\mathcal{R}$, and outputs a logit matrix $f_{\theta_r}(\mathcal{X}, \mathcal{R}) = \mathcal{P} \in \mathbb{R}^{N \times M}$, where $f$ is the decision function. Then, we apply `Softmax` function along the first dimension of $\mathcal{P}$ to obtain the probability matrix $\tilde{\mathcal{P}}$, where the entity $\tilde{\mathcal{P}}_{ij}$ represents the probability of the $i_{\text{th}}$ instance being selected for the outer minimization of the $j_{\text{th}}$ model. For each model, the neural router distributes the instances with Top$\mathcal{K}$-largest predicted probabilities as a batch for training (line 10-13). In doing so, all models have the same amount ($\mathcal{K}$) of training instances, which explicitly ensures the *load balancing* (see Appendix E). We also discuss other strategies of instance distributing, such as sampling, instance-based choice, etc. More details can be found in Section 4.2.

After the models $\Theta$ are trained with the distributed instances, we further evaluate the $N$ instances on each model, obtaining a new cost matrix $\mathcal{R}' \in \mathbb{R}^{N \times M}$. To pursue desirable *collaborative performance*, it is expected that the neural router can reasonably exploit the overall capacity of models. Since the action space is huge and the models are changing throughout the training, we resort to reinforcement learning (based on trial-and-error) to optimize parameters of the neural router $\theta_r$ (line 14-15). Specifically, we set $(\min \mathcal{R} - \min \mathcal{R}')$ as the reward signal, and update $\theta_r$ by gradient ascent to maximize the expected return with the following approximation:

$$\nabla_{\theta_r} \mathcal{L}(\theta_r | \mathcal{X}) = \mathbb{E}_{j \in (1, \dots, M), i \in \text{Top}\mathcal{K}(\tilde{\mathcal{P}}_{\cdot j}), \tilde{\mathcal{P}}}[(\min \mathcal{R} - \min \mathcal{R}')_i \nabla_{\theta_r} \log \tilde{\mathcal{P}}_{ij}], \tag{6}$$

where the $\min$ operator is applied along the last dimension of $\mathcal{R}$ and $\mathcal{R}'$, since we would like to maximize the improvement of collaborative performance after training with the selected instances. Intuitively, if an entity in $(\min \mathcal{R} - \min \mathcal{R}')$ is positive, it means that, after training with the selected instances, the collaborative performance of models on the corresponding instance is increased. Thus, the corresponding action taken by the neural router should be reinforced, and vice versa. For example, in Fig. 2, if the reward entity for the first instance $x_1$ is positive, the probability of this action (i.e., the red one in the first row of $\mathcal{P}$) will be reinforced after optimization. Note that the unselected (e.g., black ones) will be masked out in Eq. (6). In doing so, the neural router learns to effectively distribute instances that may benefit the boost of collaborative performance. More details of the model structure and the analysis of the learned routing policy are presented in Appendix C.

## 4 EXPERIMENTS

In this section, we empirically verify the effectiveness and generalizability of the proposed framework against attacks specialized for VRPs, and conduct further analyses to provide the underlying insights. Specifically, our experiments focus on two attack methods (Zhang et al., 2022; Lu et al., 2023), since the accuracy-robustness trade-off exists when conducting vanilla AT to defend against them. We conduct the main experiments on POMO (Kwon et al., 2020) with the attack generator from Zhang et al. (2022), and further demonstrate the generalizability of the proposed framework on MatNet (Kwon et al., 2021) with the attack generator from Lu et al. (2023). More details of the experimental setups, data generation and additional empirical results are presented in Appendix D.

**Baselines.** 1) *Traditional VRP methods:* we solve TSP instances by Concorde and LKH3 (Helsgaun, 2017), and CVRP instances by hybrid genetic search (HGS) (Vidal, 2022) and LKH3. 2) *Neural methods:* we compare our method with the pretrained base model POMO ($\sim$1M parameters), and its variants training with various defensive methods, such as the vanilla adversarial training (POMO_AT), the defensive method proposed by the attacker (Zhang et al., 2022) (POMO_HAC), and the diversity training (Kariyappa & Qureshi, 2019) from the literature of the adversarial robustness with ensembles (POMO_DivTrain). Specifically, POMO_AT adversarially trains the models by first generating local adversarial instances in the inner maximization, and then minimizing their empirical risks in the outer minimization. POMO_HAC further improves the outer minimization

Table 1: Performance evaluation over 1K test instances. The bracket includes the number of models.

| | | Uniform (100) Gap | Time | Fixed Adv. (100) Gap | Time | Adv. (100) Gap | Time | Uniform (200) Gap | Time | Fixed Adv. (200) Gap | Time | Adv. (200) Gap | Time |
|---|---|---|---|---|---|---|---|---|---|---|---|---|---|
| | | | | $n = 100$ | | | | | | $n = 200$ | | | |
| TSP | Concorde | 0.000% | 0.3m | 0.000% | 0.3m | – | – | 0.000% | 0.6m | 0.000% | 0.6m | – | – |
| | LKH3 | 0.000% | 1.3m | 0.002% | 2.1m | – | – | 0.000% | 3.9m | 0.005% | 5.8m | – | – |
| | POMO (1) | 0.144% | 0.1m | 35.803% | 0.1m | 35.803% | 0.1m | 0.736% | 0.5m | 63.477% | 0.5m | 63.477% | 0.5m |
| | POMO_AT (1) | 0.365% | 0.1m | 0.390% | 0.1m | 0.330% | 0.1m | 2.151% | 0.5m | 1.248% | 0.5m | 1.154% | 0.5m |
| | POMO_AT (3) | 0.255% | 0.3m | 0.295% | 0.3m | 0.243% | 0.3m | 1.884% | 1.5m | 1.090% | 1.5m | 1.011% | 1.5m |
| | POMO_HAC (3) | 0.135% | 0.3m | 0.344% | 0.3m | 0.316% | 0.3m | 0.683% | 1.5m | 1.308% | 1.5m | 1.273% | 1.5m |
| | POMO_DivTrain (3) | 0.255% | 0.3m | 0.297% | 0.3m | 0.254% | 0.3m | 1.875% | 1.5m | 1.093% | 1.5m | 1.026% | 1.5m |
| | CNF_Greedy (3) | 0.187% | 0.3m | 0.314% | 0.3m | 0.280% | 0.3m | 0.868% | 1.5m | 1.108% | 1.5m | 1.096% | 1.5m |
| | CNF (3) | **0.118%** | 0.3m | **0.236%** | 0.3m | **0.217%** | 0.3m | **0.614%** | 1.5m | **0.954%** | 1.5m | **0.952%** | 1.5m |
| CVRP | HGS | 0.000% | 6.6m | 0.000% | 14.6m | – | – | 0.000% | 0.4h | 0.000% | 1.2h | – | – |
| | LKH3 | 0.538% | 18.1m | 0.344% | 23.0m | – | – | 1.116% | 0.5h | 0.761% | 0.6h | – | – |
| | POMO (1) | 1.209% | 0.1m | 3.983% | 0.1m | 3.983% | 0.1m | 2.122% | 0.6m | 16.173% | 0.8m | 16.173% | 0.8m |
| | POMO_AT (1) | 1.456% | 0.1m | 0.882% | 0.1m | 0.935% | 0.1m | 3.249% | 0.6m | 1.384% | 0.6m | 1.435% | 0.6m |
| | POMO_AT (3) | 1.256% | 0.3m | 0.767% | 0.3m | 0.809% | 0.3m | 2.919% | 1.8m | 1.253% | 1.8m | 1.296% | 1.8m |
| | POMO_HAC (3) | 1.085% | 0.3m | 0.829% | 0.3m | 0.848% | 0.3m | 1.974% | 1.8m | 1.374% | 1.8m | 1.367% | 1.8m |
| | POMO_DivTrain (3) | 1.254% | 0.3m | 0.754% | 0.3m | 0.809% | 0.3m | 2.946% | 1.8m | 1.220% | 1.8m | 1.302% | 1.8m |
| | CNF_Greedy (3) | 1.112% | 0.3m | 0.785% | 0.3m | 0.821% | 0.3m | **1.969%** | 1.8m | 1.316% | 1.8m | 1.353% | 1.8m |
| | CNF (3) | **1.073%** | 0.3m | **0.730%** | 0.3m | **0.769%** | 0.3m | 2.031% | 1.8m | **1.193%** | 1.8m | **1.198%** | 1.8m |

by optimizing a hardness-aware instance-reweighted loss function on the mixed dataset, including both clean and local adversarial instances. POMO_DivTrain improves the ensemble diversity by minimizing the cosine similarity between the gradients of models w.r.t. the input. We further compare our method with CNF_Greedy by replacing the neural router with the heuristic greedy selection method. More implementation details of baselines are provided in Appendix D.1.

**Training Setups.** CNF starts with a pretrained model, and then adversarially trains its $M$ copies in a collaborative way. We consider two scales of training instances $n \in \{100, 200\}$. For the pretraining stage, the model is trained on clean instances following the uniform distribution. We use the open-source pretrained models[1] for $n = 100$, and retrain the models for $n = 200$. Following the original training setups from Kwon et al. (2020), Adam optimizer (Kingma & Ba, 2015) is used with the learning rate of $1e - 4$, the weight decay of $1e - 6$ and the batch size of $B = 64$. To achieve full convergence, we pretrain the models on 300M and 100M clean instances for TSP200 and CVRP200, respectively. After obtaining the pretrained model, we use it to initialize $M = 3$ models, and further adversarially train them on 5M and 2.5M instances for $n = 100$ and $n = 200$, respectively. To save the GPU memory, we reduce the batch size to $B = 32$ for $n = 200$. The optimizer setting is the same as the one in the pretraining stage, except that the learning rate is decayed by 10 for the last 40% training instances. For the mixed data collection, we collect $B$ clean instances, $MB$ local adversarial instances and $B$ global adversarial instances in each training step.

**Inference Setups.** For neural methods, we use the greedy rollout with x8 instance augmentations following Kwon et al. (2020). We report the average gap over the test dataset containing 1K instances. Concretely, the gap is computed w.r.t. the traditional VRP solvers (i.e., Concorde for TSP, and HGS for CVRP). If multiple models exist, we report the collaborative performance, where the best gap among all models is recorded for each instance. The reported time is the total time to solve the entire dataset. We consider three evaluation metrics: 1) *Uniform (standard generalization):* the performance on clean instances whose distributions are the same as the pretraining ones; 2) *Fixed Adv. (adversarial robustness):* the performance on adversarial instances generated by attacking the pretrained model. It mimics the black-box setting, where the attacker generates adversarial instances using a surrogate model due to the inaccessibility to the current model and the transferability of adversarial instances; 3) *Adv. (adversarial robustness):* the performance on adversarial instances generated by attacking the current model. For methods with multiple ($M$) models, it generates $MK$ adversarial instances, from which we randomly sample 1K instances to construct the test dataset. It is the conventional white-box metric used to evaluate adversarial robustness in the literature of AT.

## 4.1 PERFORMANCE EVALUATION

The results are shown in Table 1. We have conducted t-test with the threshold of 5% to verify the statistical significance. The results of traditional VRP methods on the metric of Adv. is not shown since the generated adversarial dataset is different for each neural method. For all neural methods, we report the inference time on a single GPU. Moreover, for neural methods with multiple

---

[1] https://github.com/yd-kwon/POMO

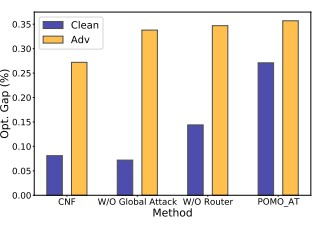
(a) Ablation study on Components

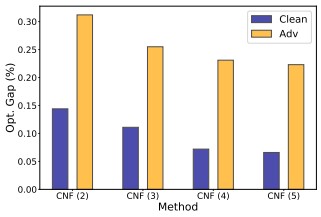
(b) Ablation study on Hyperparameters

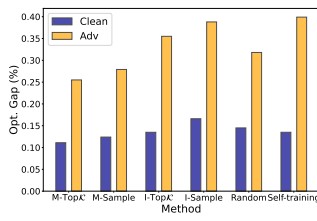
(c) Ablation study on Routing Strategies

Figure 3: Ablation studies on TSP100. The metrics of *Uniform* and *Fixed Adv.* are reported.

($M$) models (e.g., POMO_AT (3)), we further develop an implementation of parallel evaluation on multiple GPUs, which could reduce their inference time by almost $M$ times. From the results, we observe that 1) traditional VRP methods are relatively more robust than the neural methods against crafted perturbations, demonstrating the importance and necessity of improving adversarial robustness for neural methods; 2) the evaluation metrics of Fixed Adv. and Adv. are almost consistent in the context of VRPs; 3) our method consistently outperforms baselines, and achieves high standard generalization and adversarial robustness concurrently. For our method, we further give an example about the performance of each model on TSP100 as shown in Fig. 5. Although not all models perform well on both clean and adversarial instances, the collaborative performance is quite good, demonstrating the capability of CNF in reasonably exploiting the overall capacity of models.

## 4.2 ABLATION STUDY

We conduct extensive ablation studies on TSP100 to demonstrate the effectiveness and sensitivity of our method. Note that the setups are slightly different from the training ones (e.g., half the training instances). The detailed results and setups are presented in Fig. 3 and Appendix D.1, respectively.

**Ablation on Components.** We investigate the role of each component in our method by removing them separately. As demonstrated in Fig. 3(a), despite both components contribute to the collaborative performance, the neural router exhibits a bigger effect due to its favorable potentials to elegantly exploit training instances and model capacity, especially in the presence of multiple models.

**Ablation on Hyperparameters.** We investigate the effect of the number of trained models, which is a key hyperparameter of our method, with respect to the collaborative performance. The results are shown in Fig. 3(b), where we observe that increasing the number of models could further improve the collaborative performance. However, we use $M = 3$ in the main experiments due to the trade-off between performance and computational complexity. We refer to Appendix D.5 for more results.

**Ablation on Routing Strategies.** We further discuss different routing strategies, including neural and heuristic ones. Specifically, given the logit matrix $\mathcal{P}$ predicted by the neural router, there are various ways to distribute the instances: 1) *Model choice with Top$\mathcal{K}$ (M-Top$\mathcal{K}$):* each model chooses potential instances with Top$\mathcal{K}$-largest logits, which is the default strategy ($\mathcal{K} = B$) in CNF; 2) *Model choice with sampling (M-Sample):* each model chooses potential instances by sampling from the probality distribution (i.e., scaled logits); 3/4) *Instance choice with Top$\mathcal{K}$/sampling (I-Top$\mathcal{K}$/I-Sample):* in contrast to the model choice, each instance chooses potential model(s) either by Top$\mathcal{K}$ or sampling. The probability matrix $\tilde{\mathcal{P}}$ is obtained by taking Softmax along the first and last dimension of $\mathcal{P}$ for model choice and instance choice, respectively. Unlike the model choice, instance choice cannot guarantee load balancing. For example, the majority of instances may choose a dominant model (if exists), leaving the remaining models underfitting and therefore weakening the collaborative performance; 5) *Random:* the instances are randomly distributed to each model; 6) *Self-training:* each model is trained on the adversarial instances generated by itself. The results in Fig. 3(c) show that M-Top$\mathcal{K}$ performs the best among all strategies. Moreover, as discussed above, we empirically observe that the dominant model tends to appear when using I-Top$\mathcal{K}$ or I-Sample.

## 4.3 OUT-OF-DISTRIBUTION GENERALIZATION

In contrast to other domains, the set of valid problems is not just a low-dimensional manifold in a high-dimensional space, and hence the manifold hypothesis (Stutz et al., 2019) does not apply to VRPs (or COPs). Therefore, it is critical for neural methods to perform well on adversarial instances

Table 2: Generalization evaluation on synthetic TSP datasets. The model is only trained on $n$=100.

| | Cross-Distribution | | | | Cross-Size | | | | Cross-Size & Distribution | | | |
| | Rotation (100) | | Explosion (100) | | Uniform (50) | | Uniform (200) | | Rotation (200) | | Explosion (200) | |
| | Gap | Time | Gap | Time | Gap | Time | Gap | Time | Gap | Time | Gap | Time |
|---|---|---|---|---|---|---|---|---|---|---|---|---|
| Concorde | 0.000% | 0.3m | 0.000% | 0.3m | 0.000% | 0.2m | 0.000% | 0.6m | 0.000% | 0.6m | 0.000% | 0.6m |
| LKH3 | 0.000% | 1.2m | 0.000% | 1.2m | 0.000% | 0.4m | 0.000% | 3.9m | 0.000% | 3.3m | 0.000% | 3.5m |
| POMO (1) | 0.471% | 0.1m | 0.238% | 0.1m | 0.064% | 0.1m | 1.658% | 0.5m | 2.936% | 0.5m | 2.587% | 0.5m |
| POMO_AT (1) | 0.640% | 0.1m | 0.364% | 0.1m | 0.151% | 0.1m | 2.667% | 0.5m | 3.462% | 0.5m | 2.989% | 0.5m |
| POMO_AT (3) | 0.508% | 0.3m | 0.263% | 0.3m | 0.085% | 0.1m | 2.362% | 1.5m | 3.176% | 1.5m | 2.688% | 1.5m |
| POMO_HAC (3) | 0.204% | 0.3m | 0.107% | 0.3m | 0.038% | 0.1m | 1.414% | 1.5m | 2.184% | 1.5m | 1.718% | 1.5m |
| POMO_DivTrain (3) | 0.502% | 0.3m | 0.255% | 0.3m | 0.078% | 0.1m | 2.356% | 1.5m | 3.176% | 1.5m | 2.707% | 1.5m |
| CNF (3) | **0.193%** | 0.3m | **0.084%** | 0.3m | **0.036%** | 0.1m | **1.383%** | 1.5m | **2.055%** | 1.5m | **1.672%** | 1.5m |

when striving for a broader Out-of-Distriburion (OOD) generalization in VRPs (see Appendix E). In this section, we further evaluate the OOD generalization performance on unseen instances from both synthetic and benchmark datasets. The empirical results demonstrate that raising robustness against adversarial instances by CNF favorably promotes various types of generalization, indicating the potential existence of neural VRP solvers with high generalization and robustness concurrently. The data generation and full results could be found in Appendix D.2 and D.3, respectively.

**Synthetic Datasets.** We consider three generalization settings, i.e., cross-distribution, cross-size, and cross-size & distribution. The results are shown in Table 2, from which we observe that simply conducting the vanilla AT somewhat hurts the OOD generalization, while our method could significantly improve it. Since adversarial robustness is known as a kind of local generalization property (Goodfellow et al., 2015; Madry et al., 2018), the improvements in OOD generalization could be viewed as a byproduct of defending against adversarial instances and balancing the trade-off.

**Benchmark Datasets.** We further evaluate all methods on the real-world benchmark instances, such as TSPLIB (Reinelt, 1991) and CVRPLIB (including Set-X (Uchoa et al., 2017) and Set-XML100 (Queiroga et al., 2022)). The results are shown in Appendix D.3, where we observe that our method performs well on most of the instances. We also present the results of Omni-VRP (Zhou et al., 2023), which is a recent work leveraging the meta-learning techniques (Finn et al., 2017; Nichol et al., 2018) to directly deal with the OOD generalization issue of neural VRP methods. Note that its results are only for reference due to different problem settings.

## 4.4 VERSATILITY

To demonstrate the generalizability of CNF, we further evaluate it against another attack method (Lu et al., 2023). Specifically, they attack MatNet (Kwon et al., 2021) by lowering the cost of a partial asymmetric TSP (ATSP) instance. When simply following the vanilla AT, the undesirable trade-off could be also observed in the empirical results of Lu et al. (2023), while our method is able to train models with both high standard generalization and adversarial robustness. The detailed attack method, training setups, and results are presented in Appendix B.3, D.1 and D.4, respectively.

## 5 CONCLUSION

This paper studies the crucial yet underexplored adversarial defense of neural VRP methods. We propose a collaborative neural framework (CNF) to adversarially train multiple models in a collaborative manner, which could achieve high standard generalization and adversarial robustness simultaneously. Our work demonstrates the favorable potential of defending against adversarial instances in promoting various types of generalization of neural VRP methods. We hope our insights could help the community to build a more robust and generalizable neural VRP solver. The limitation of this work is the increased computational complexity due to the need of synergistically training multiple models. Fortunately, based on the empirical results, the CNF with 3 models have already achieved decent performance. It could even outperform the vanilla AT trained with 5 models, demonstrating a better trade-off between performance and computational complexity. Interesting future research directions may include: 1) designing efficient and effective attack methods for other COPs; 2) pursuing better adversarial robustness with fewer computation resources, such as the conditional computation (Jacobs et al., 1991; Shazeer et al., 2017) and sharing parameters (Xin et al., 2021a) for the encoder-decoder-based architecture; 3) explicitly considering OOD generalization when dealing with adversarial robustness, or vice versa; and 4) investigating whether large language models (Yang et al., 2023) are able to (robustly) approximate optimal solutions to COPs.

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

REPRODUCIBILITY STATEMENT

We promise that the source code, test datasets (i.e., instances and optimal solutions), and pretrained models will be made public for research reproducibility. The experimental setups for training and evaluation as well as the hyperparamters are detailedly described in Section 4 and Appendix D.

APPENDIX

The Appendix is organized as follows. In Appendix A, we give a comprehensive review of the related works. In Appendix B, we give an introduction to the attack methods for vehicle routing problems (VRPs). In Appendix C, we present the details and further analyses about the neural router. In Appendix D, we describe the experimental setups, data generation and additional empirical results. In Appendix E, we conclude our paper with further discussions.

## A    RELATED WORK

**Neural Methods for VRPs.** Most neural methods for VRPs learn construction heuristics, which are mainly divided into two categories, i.e., autoregressive and non-autoregressive ones. Autoregressive methods sequentially construct the solution by adding one feasible node at a step. Vinyals et al. (2015) proposes the Pointer Network (Ptr-Net) to solve TSP with supervised learning. Subsequent works train Ptr-Net with reinforcement learning (RL) to solve TSP (Bello et al., 2017) and CVRP (Nazari et al., 2018). Kool et al. (2018) introduces the attention model (AM) based on the Transformer architecture (Vaswani et al., 2017) to solve a wide range of COPs including TSP and CVRP. Kwon et al. (2020) further proposes the policy optimization with multiple optima (POMO), which improves upon AM by exploiting solution symmetries. Regarding non-autoregressive methods, the solution is constructed in a one-shot manner, without iterative forward passes through the model (decoder). Joshi et al. (2019) leverages the graph convolutional network to predict the probability of each edge appearing on the optimal tour (i.e., heat-map) using supervised learning. Recent works (Fu et al., 2021; Qiu et al., 2022; Sun & Yang, 2023) further improve its performance and scalability by using advanced training paradigms (e.g., RL), search strategies (e.g., active search (Bello et al., 2017; Hottung et al., 2022) and Monte-Carlo Tree Search (Silver et al., 2016)) and models (e.g., diffusion model (Sohl-Dickstein et al., 2015; Ho et al., 2020)). We refer to Ma et al. (2019); Kwon et al. (2021); Xin et al. (2021a); Kim et al. (2022); Choo et al. (2022); Grinsztajn et al. (2022) for further advances on learning construction heuristics, and to Dai et al. (2017); Selsam et al. (2019); Yolcu & Póczos (2019); Ahn et al. (2020) for other COPs.

On the other hand, some neural methods learn improvement heuristics to refine an initial feasible solution iteratively, until a termination condition is satisfied. In this line of research, the classic local search methods (e.g., 2-opt (Croes, 1958), large neighborhood search (LNS) (Shaw, 1998)) and specialized heuristic solvers for VRPs (e.g., Lin-Kernighan-Helsgaun (LKH) (Helsgaun, 2000; 2017)) are usually exploited (Chen & Tian, 2019; Lu et al., 2020; Hottung & Tierney, 2020; d O Costa et al., 2020; Wu et al., 2021; Wang et al., 2021; Ma et al., 2021; Xin et al., 2021b; Kim et al., 2021; Hudson et al., 2022). In general, the improvement heuristics could achieve better performance than the construction ones, but at the expense of much longer inference time.

**Robustness of Neural Methods.** There is a recent research trend on the robustness of neural methods for COPs (Varma & Yoshida, 2021; Geisler et al., 2022; Lu et al., 2023), with only a few works on VRPs (Zhang et al., 2022; Geisler et al., 2022; Lu et al., 2023). In general, they primarily focus on *attacking neural construction heuristics* by introducing effective ways to generate adversarial instances that are underperformed by the current model. Following the AT paradigm, Zhang et al. (2022) perturbs node coordinates of TSP instances by solving an inner maximization problem (similar to the fast gradient sign method (FGSM) (Goodfellow et al., 2015)), and trains the model with an instance-reweighted loss function. Geisler et al. (2022) proposes an efficient and sound perturbation model, which ensures the optimal solution to the perturbed TSP instance could be easily derived. It adversarially inserts several nodes into the clean instance by maximizing the cross-entropy over the edges, so that the predicted route is maximally different from the derived optimal one. Lu et al. (2023) leverages a no-worse optimal cost guarantee (i.e., by lowering the cost of a partial problem) to generate adversarial instances for asymmetric TSP. However, existing methods mainly follow the

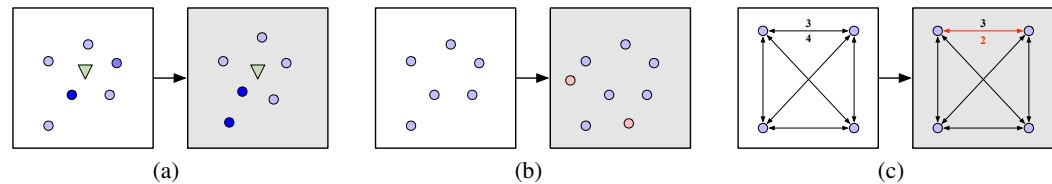

(a)          (b)          (c)

Figure 4: An illustration of generated adversarial instances (i.e., the grey ones). (a) An adversarial instance generated by Zhang et al. (2022) on CVRP, where the triangle represents the depot node. A deeper color denotes a heavier node demand; (b) An adversarial instance generated by Geisler et al. (2022) on TSP, where the red nodes represent the inserted adversarial nodes; (c) An adversarial instance generated by Lu et al. (2023) on ATSP, where the cost of an edge is in half.

vanilla AT (Madry et al., 2018) to deploy the defense, leaving a considerable gap to further consolidate the robustness. More advanced AT variants, such as TRADES (Zhang et al., 2019), AT with ensemble (Tramèr et al., 2018; Pang et al., 2019; Yang et al., 2020a), and AT in RL (Pinto et al., 2017) may be of interest, but it is non-trivial to adapt them to the discrete COP domain due to their dependence on the imperceptible perturbation model (see Appendix E). In this paper, we specialize a general collaborative neural framework to learn robust neural methods for VRPs.

## B   ATTACK METHODS

In this section, we first give a definition of the adversarial instance in VRPs, and then present the details of recently proposed attack methods for neural VRP solvers, including perturbing input attributes (Zhang et al., 2022), adversarially inserting new nodes (Geisler et al., 2022), and lowering the cost of a partial problem instance to ensure no-worse theoretical optimum (Lu et al., 2023). They could generate instances that are underperformed by the current model.

We define the adversarial instance as the instance that 1) is obtained by the perturbation model $G$ within the neighborhood of the clean instance, and 2) is underperformed by the current model. Formally, given a clean VRP instance $x = \{x_c, x_d\}$, where $x_c \in \mathcal{N}_c$ is the continuous attribute (e.g., node coordinates) within the valid range $\mathcal{N}_c$, and $x_d \in \mathcal{N}_d$ is the discrete attribute (e.g., node demand) within the valid range $\mathcal{N}_d$, the adversarial instance $\tilde{x}$ is found by the perturbation model $G$ around the clean instance $x$, on which the current model may be vulnerable. Technically, the adversarial instance $\tilde{x}$ can be constructed by adding crafted perturbations $\gamma_{x_c}, \gamma_{x_d}$ to the corresponding attribute of the clean instance, and then project them back to the valid domain: $\tilde{x}_c = \Pi_{\mathcal{N}_c}(x_c + \alpha \cdot \gamma_{x_c})$, $\tilde{x}_d = \Pi_{\mathcal{N}_d}(x_d + \alpha \cdot \gamma_{x_d})$, where $\Pi$ denotes the projection operator (e.g., min-max normalization), and $\alpha$ denotes the attack budget. The crafted perturbations could be obtained by various perturbation models, such as the one in Eq. 7, e.g., $\gamma_c = \nabla_{x_c} \ell(x; \theta)$, $\gamma_d = \nabla_{x_d} \ell(x; \theta)$, where $\theta$ is the model parameters, $\ell$ is the loss function. We omit the attack step $t$ for notation simplicity. An illustration of the adversarial instance generated by each attack method is shown in Fig. 4. Below, we follow the notations from the main paper, and introduce each attack method in details.

### B.1   ATTRIBUTE PERTURBATION

The attack generator from Zhang et al. (2022) is applied to attention-based models (Kool et al., 2018; Kwon et al., 2020) by perturbing attributes (e.g., node coordinates) of input instances. As introduced in Section 3, it generates adversarial instances by directly maximizing the reinforcement loss variant (so called the hardness measure in Zhang et al. (2022))). We take the perturbation on the node coordinate as an example. Suppose given the clean instance $x$ (i.e., $\tilde{x}^{(0)} = x$) and model parameter $\theta$, the solution to the inner maximization could be approximated as follows:

$$\tilde{x}^{(t+1)} = \Pi_{\mathcal{N}_c}[\tilde{x}^{(t)} + \alpha \cdot \nabla_{\tilde{x}^{(t)}} \ell(\tilde{x}^{(t)}; \theta^{(t)})], \tag{7}$$

where $\tilde{x}^{(t)}$ is the (local) adversarial instances and $\theta^{(t)}$ is the model parameters, at step $t$. Here we use $\mathcal{N}_c$ to represent the valid domain of node coordinates (i.e., unit square $U(0, 1)$) for simplicity. After each iteration, it checks whether $\hat{x}^{(t)} = \tilde{x}^{(t)} + \alpha \cdot \nabla_{\tilde{x}^{(t)}} \ell(\tilde{x}^{(t)}; \theta^{(t)})$ is within the valid domain $\mathcal{N}_c$ or not. If it is out of $\mathcal{N}_c$, the projection operator (i.e., min-max normalization) is applied as follows:

$$\Pi_{\mathcal{N}_c}(\hat{x}^{(t)}) = \frac{\hat{x}^{(t)} - \min \hat{x}^{(t)}}{\max \hat{x}^{(t)} - \min \hat{x}^{(t)}}(\max \mathcal{N}_c - \min \mathcal{N}_c). \tag{8}$$

Note that it originally only focuses on TSP, where the node coordinates are perturbed. We further adapt it to CVRP by perturbing both the node coordinates and node demands. The implementation is straightforward, except that we set the valid domain of node demands as $\mathcal{N}_d = \{1, \ldots, 9\}$. For the perturbations on node demands, the projection operator applies another round operation as follows:

$$\Pi_{\mathcal{N}_d}(\hat{x}^{(t)}) = \lceil \frac{\hat{x}^{(t)} - \min \hat{x}^{(t)}}{\max \hat{x}^{(t)} - \min \hat{x}^{(t)}} (\max \mathcal{N}_d - \min \mathcal{N}_d) \rceil. \tag{9}$$

### B.2 NODE INSERTION

An efficient and sound perturbation model is proposed by Geisler et al. (2022), which, given the optimal solution $y$ to the clean instance $x$ sampled from the data distribution $\mathcal{D}$, guarantees to directly derive the optimal solution $\tilde{y}$ to the adversarial instance $\tilde{x}$ without running a solver. The attack is applied to the GCN (Joshi et al., 2019), which is a non-autoregressive construction method for TSP. It learns the probability of each edge occurring in the optimal solution (i.e., heat-map) with supervised learning. Following the AT framework, the objective function could be written as follows:

$$\min_{\theta} \mathbb{E}_{(x,y) \sim \mathcal{D}} \max_{\tilde{x}} \ell(f_{\theta}(\tilde{x}), \tilde{y}), \text{ with } \tilde{x} \in G(x, y) \wedge \tilde{y} = h(\tilde{x}, x, y), \tag{10}$$

where $\ell$ is the cross-entropy loss; $G$ is the perturbation model that describes the possible perturbed instances $\tilde{x}$ around the clean instance $x$; and $h$ is used to derive the optimal solution $\tilde{y}$ based on $(\tilde{x}, x, y)$ without running a solver. In the inner maximization, the adversarial instance $\tilde{x}$ is generated by inserting several new nodes into $x$ (below we take inserting one new node as an example), which adheres to below proposition and proof (by contradiction) borrowed from Geisler et al. (2022):

**Proposition 1.** *Let $Z \notin \mathcal{V}$ be an additional node to be inserted, $w(\mathcal{E})$ is an edge weight, and $P, Q$ are any two neighbouring nodes in the original optimal solution $y$. Then, the new optimal solution $\tilde{y}$ (including $Z$) is obtained from $y$ through inserting $Z$ between $P$ and $Q$ if $\nexists(A, B) \in \mathcal{E} \setminus \{(P, Q)\}$ with $A \neq B$ s.t. $w(A, Z) + w(B, Z) - w(A, B) \leq w(P, Z) + w(Q, Z) - w(P, Q)$.*

**Proof.** Let $(R, S) \in \mathcal{E} \setminus \{(P, Q)\}$ to be two neighboring nodes of $Z$ on $\tilde{y}$. Assume $w(P, Z) + w(Q, Z) - w(P, Q) < w(R, Z) + w(S, Z) - w(R, S)$ and the edges $(P, Z)$ and $(Q, Z)$ are not contained in $\tilde{y}$ (i.e., $Z$ is inserted between $R$ and $S$ rather than $P$ and $Q$).

Below inequalities hold by the optimality of $y$ and $\tilde{y}$:

$$c(\tilde{y}) - w(R, Z) - w(S, Z) + w(R, S) \geq c(y). \tag{11}$$

$$c(y) + w(P, Z) + w(Q, Z) - w(P, Q) \geq c(\tilde{y}). \tag{12}$$

Therefore, we have

$$c(y) + w(P, Z) + w(Q, Z) - w(P, Q) \geq c(\tilde{y}) \geq c(y) + w(R, Z) + w(S, Z) - w(R, S), \tag{13}$$

which leads to a contradiction against the assumption (i.e., $w(P, Z) + w(Q, Z) - w(P, Q) < w(R, Z) + w(S, Z) - w(R, S)$). The proof is completed.

They use a stricter condition $\nexists(A, B) \in \mathcal{E} \setminus \{(P, Q)\}$ with $A \neq B$ s.t. $w(A, Z) + w(B, Z) - w(A, B) \leq w(P, Z) + w(Q, Z) - w(P, Q)$ in the proposition, since it is unknown which nodes could be $R$ and $S$ in $\tilde{y}$. Moreover, for the metric TSP, whose node coordinate system obeys the triangle inequality (e.g., euclidean distance), it is sufficient if the condition of Proposition 1 holds for $(A, B) \in \mathcal{E} \setminus (\{(P, Q)\} \cup \mathcal{H})$ with $A \neq B$ where $\mathcal{H}$ denotes the pairs of nodes both on the Convex Hull $\mathcal{H} \in CH(\mathcal{E})$ that are not a line segment of the Convex Hull. It is due to the fact that the optimal route $\tilde{y}$ must be a simple polygon (i.e., no crossings are allowed) in the metric space. This conclusion was first stated for the euclidean space as "the intersection theorem" by Cutler (1980) and is a direct consequence of the triangle inequality.

Based on the above-mentioned proposition, the optimization of inner maximization involves: 1) obtaining the coordinates of additional node $Z$ by gradient ascending (e.g., maximizing $l$ such that the model prediction is maximally different from the derived optimal solution $\tilde{y}$); 2) penalizing if $Z$ violates the constraint in Proposition 1. Unfortunately, the constraint is non-convex and hard to find a relaxation. Instead of optimizing the Lagrangian (which requires extra computation for evaluating $f_{\theta}$), the vanilla gradient descent is leveraged with the constraint as the objective:

$$Z \leftarrow Z - \eta \nabla_Z [w(P, Z) + w(Q, Z) - w(P, Q) - (\min_{A, B} w(A, Z) + w(B, Z) - w(A, B))], \tag{14}$$

where $\eta$ is the step size. After we find $Z$ satisfying the constraint, the adversarial instance $\tilde{x}$ and the corresponding optimal solution $\tilde{y}$ could be constructed directly. Finally, the outer minimization takes $(\tilde{x}, \tilde{y})$ as inputs to train the robust neural solvers. This attack method could be easily adopted by our proposed framework, where $\theta$ is replaced by the best model $\theta_b$ when maximizing the cross-entropy loss (i.e., $\max_{\tilde{x}} \ell(f_{\theta_b}(\tilde{x}), \tilde{y})$) in the inner maximization optimization. However, due to the efficiency and soundness of the perturbation model, it does not suffer from the undesirable trade-off following the vanilla AT (Geisler et al., 2022). Therefore, we mainly focus on other attack methods (Zhang et al., 2022; Lu et al., 2023) in our experiments.

### B.3 NO-WORSE THEORETICAL OPTIMUM

The attack method specialized for graph-based COPs is proposed by Lu et al. (2023). It resorts to the black-box adversarial attack method by training a reinforcement learning based attacker, and therefore could be used to generate adversarial instances for both differentiable (e.g., learning-based) and non-differentiable (e.g., heuristic or exact) solvers. In this paper, we only consider the learning-based neural solvers for VRPs. Specifically, it generates the adversarial instance $\tilde{x}$ by modifying the clean instance $x$ (e.g., lowering the cost of a partial instance) under the no worse optimum condition, which requires $c(\tilde{y}) \leq c(y)$ if we are solving a minimization optimization problem. The attack is successful (w.r.t. the neural solver $\theta$) if the output solution to $\tilde{x}$ is worse than the one to $x$ (i.e., $c(\tilde{\tau}|\tilde{x}; \theta) > c(\tau|x; \theta) \geq c(y) \geq c(\tilde{y})$). The training of the attacker is hence formulated as follows:

$$\max_{\tilde{x}}. \quad c(\tilde{\tau}|\tilde{x}; \theta) - c(\tau|x; \theta),$$
$$\text{s.t.} \quad \tilde{x} = G(x, T; \theta), \ c(\tilde{y}) \leq c(y), \tag{15}$$

where $G(x, T; \theta)$ represents the deployment of the attacker $G$ trained on the given model (or solver) $\theta$ to conduct $T$ modifications on the clean instance $x$. It is trained with the objective as in Eq. (15) using the RL algorithm (i.e., Proximal Policy Optimization (PPO) (Schulman et al., 2017)). Specifically, the attack process is modelled as a MDP, where, at step $t$, the state is the current instance $\tilde{x}^{(t)}$; the action is to select an edge whose weight is going to be half; and the reward is the increase of the objective: $c(\tau^{(t+1)}|\tilde{x}^{(t+1)}; \theta) - c(\tau^{(t)}|\tilde{x}^{(t)}; \theta)$. This process is iterative until $T$ edges are modified. We use ROCO to represent this attack method in the remaining of this paper.

ROCO has been applied to attack MatNet (Kwon et al., 2021) on asymmetric TSP (ATSP). As shown in the empirical results of Lu et al. (2023), conducting the vanilla AT may suffer from the trade-off between standard generalization and adversarial robustness. To solve the problem, we further apply our method to defend against it. However, it is not straightforward to adapt ROCO to the inner maximization of CNF, since ROCO belongs to the black-box adversarial attack method, which does not directly rely on the parameters (or gradients) of the current model to generate adversarial instances. Concretely, we first train an attacker $G_j$ using RL for each model $\theta_j$, obtaining $M$ attackers for $M$ models ($\Theta = \{\theta_j\}_{j=0}^{M-1}$) in CNF. For the local attack, we simply use $G_j$ to generate local adversarial instances for $\theta_j$ by $G_j(x, T; \theta_j)$. For the global attack, we decompose the generation process (i.e., modifying $T$ edges) of a global adversarial instance $\bar{x}$ as follows:

$$\bar{x}^{(t+1)} = G_b^{(t)}(\bar{x}^{(t)}, 1; \theta_b^{(t)}), \quad \theta_b^{(t)} = \arg\min_{\theta \in \Theta} c(\tau|\bar{x}^{(t)}; \theta), \tag{16}$$

where $\bar{x}^{(t)}$ is the global adversarial instance; $\theta_b^{(t)}$ is the best-performing model (w.r.t. $\bar{x}^{(t)}$); and $G_b^{(t)}$ is the attacker corresponding to $\theta_b^{(t)}$, at step $t \in [0, T-1]$. Since the model $\theta_j$ is updated throughout the optimization, to save the computation, we fix the attacker $G_j$ and only update (by retraining) it every $E$ epochs using the latest model. More details and results could be found in Appendix D.4.

## C NEURAL ROUTER

In this section, we present the model structure of the neural router and the learned routing policy.

### C.1 MODEL STRUCTURE

Without loss of generality, we take the TSP as an example, where an instance consists of coordinates of $n$ nodes. The attention-based neural router takes as inputs $N$ instances $X \in \mathbb{R}^{N \times n \times 2}$ and the

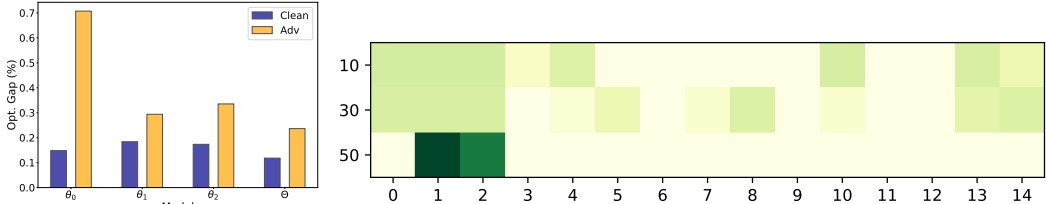

Figure 5: *Left panel:* Performance of each model $\theta_j$ and the overall collaboration performance of $\Theta$, with $M = 3$. *Right panel:* A demonstration (i.e., attention map) of the learned routing policy for the model $\theta_0$. The horizontal axis is the index of the training instance. Concretely, 0-2: clean instances $x$; 3-11: local adversarial instances $\tilde{x}$; 12-14: global adversarial instances $\bar{x}$. The vertical axis is the epoch of the checkpoint. A deeper color represents a higher probability to be chosen.

cost matrix $\mathcal{R} \in \mathbb{R}^{N \times M}$, where $M$ is the number of models in CNF. The neural router first embeds the raw inputs into h-dimensional (i.e., 128) features as follows:

$$F_I = \texttt{Mean}(W_1 X + b_1), \quad F_R = W_2 \mathcal{R} + b_2, \tag{17}$$

where $F_I \in \mathbb{R}^{N \times h}$ and $F_R \in \mathbb{R}^{N \times h}$ are features of instances and cost matrices, respectively; $W_1, W_2$ are weight matrices; $b_1, b_2$ are biases; and the $\texttt{Mean}$ operator is taken along the second dimension of inputs. Then, a single-head attention layer (i.e., *glimpse* (Bello et al., 2017)) is applied:

$$Q = W_Q([F_I, F_R]), \quad K = W_K(\texttt{Emb}(M)), \tag{18}$$

where $[\cdot, \cdot]$ is the horizontal concatenation operator; $Q \in \mathbb{R}^{N \times h}$ is the query matrix; $K \in \mathbb{R}^{M \times h}$ is the key matrix; $W_Q, W_K$ are the weight matrices; and $\texttt{Emb}(M) \in \mathbb{R}^{M \times h}$ is a learnable embedding layer representing the features of $M$ models. The logit matrix $\mathcal{P} \in \mathbb{R}^{N \times M}$ is calculated as follows:

$$\mathcal{P} = C \cdot \texttt{tanh}(\frac{QK^T}{\sqrt{h}}), \tag{19}$$

where the result is clipped by the $\texttt{tanh}$ function with $C = 10$ following Bello et al. (2017). When the neural router is applied to CVRP, we only slightly modify $Q$ by further concatenating it with the features of the depot and node demands, while keeping others the same.

### C.2 LEARNED ROUTING POLICY

Below, we try to briefly interpret the learned routing policy. We first show the performance of each model $\theta_j \in \Theta$ in the left panel of Fig. 5, which is trained on TSP100 following the training setups described in Section 4. Although not all models perform well on both clean and adversarial instances, the collaborative performance of $\Theta$ is quite good, demonstrating the capability of CNF in reasonably exploiting the overall capacity of models. We further give a demonstration (i.e., attention map) of the learned policy in the right panel of Fig. 5. We take the first model $\theta_0$ as an example, which performs well on clean instances. For simplicity and readability of the results, the batch size is set to $B = 3$, and therefore the number of input instances $\mathcal{X}$ into the neural router is 15 (see Section 3 and 4). The neural router then distributes $B = 3$ instances to each model (if with model choice routing strategies). Note that the instances for different epochs are not the same, while the types remain the same (e.g., instances with ids 0-2 are clean instances). From the results, we observe that 1) the learned policy tends to distribute clean instances and (local and global) adversarial instances to the selected model at the beginning of training, when the model is vulnerable to adversarial instances; 2) clean instances are likely to be selected at the end of training, when the model is relatively robust to adversarial instances while trying to mitigate the trade-off.

## D EXPERIMENTS

### D.1 EXTRA SETUPS

**Setups for Baselines.** All experiments are conducted on a machine with NVIDIA V100S-PCIE cards and AMD EPYC 7513 CPU at 2.6GHz. As shown in Section 4, we compare our method with

several strong traditional and neural VRP methods. Following the conventional setups in the community (Kool et al., 2018; Kwon et al., 2020; Hottung et al., 2022), for specialized heuristic solvers such as Concorde, LKH3 and HGS, we run them on 32 CPU cores for solving TSP and CVRP instances, while running neural methods on one GPU card. Below, we provide the implementation details of baselines. 1) Concorde: We use Concorde[2] Version 03.12.19 with the default setting, to solve TSP instances. 2) LKH3 (Helsgaun, 2017): We use LKH3[3] Version 3.0.8 to solve TSP and CVRP instances. For each instance, we run LKH3 with 10000 trails and 1 run. 3) HGS (Vidal, 2022): We run HGS[4] with the default hyperparameters to solve CVRP instances. The maximum number of iterations without improvement is set to 20000. 4) For POMO (Kwon et al., 2020), in addition to the open-source pretrained model, we further train it using the vanilla AT framework (POMO_AT). Specifically, following the training setups as presented in Section 4, we use the pretrained model to initialize $M$ models, and train them individually using local adversarial instances generated by each model. 5) POMO_HAC (Zhang et al., 2022) further improves upon the vanilla AT. It constructs a mixed dataset with both local adversarial instances and clean instances for training afterwards. In the outer minimization, it optimizes an instance-reweighted loss function based on the instance hardness. Following their setups, the weight for each instance $x_i$ is defined as: $w_i = \exp(\mathcal{F}(\mathcal{H}(x_i))/\mathcal{T})/\sum_j \exp(\mathcal{F}(\mathcal{H}(x_j))/\mathcal{T})$, where $\mathcal{F}$ is the transformation function (i.e., tanh). $\mathcal{T}$ is the temperature controlling the weight distribution. It starts from 20 and decreases linearly as the epoch increasing. The hardness $\mathcal{H}$ is computed the same as Eq. (4). 6) POMO_DivTrain is adapted from the diversity training (Kariyappa & Qureshi, 2019), which studies the adversarial robustness of multiple models (e.g., ensembles) in the image domain by proposing a novel method to train an ensemble of models with uncorrelated loss functions. Specifically, it improves the ensemble diversity by minimizing the cosine similarity between the gradients of (sub-)models w.r.t. the input. Its loss function is formulated as: $\mathcal{L} = \ell + \lambda \log(\sum_{1 \leq a < b \leq M} \exp(\frac{<\nabla_x \ell_a, \nabla_x \ell_b>}{|\nabla_x \ell_a||\nabla_x \ell_b|}))$, where $\ell$ is the original loss function, $M$ is the number of models, $\nabla_x \ell_a$ is the gradient of the loss function (on the $a_{th}$ model) w.r.t. the input $x$, and $\lambda = 0.5$ is a hyperparameter controlling the importance of gradient alignment during training. 7) CNF_Greedy: the neural router is simply replaced by the heuristic method, where each model selects $\mathcal{K}$ hardest instances. We use Zhang et al. (2022) as the attack method in Section 4. For simplicity and training efficiency, we only perturb node coordinates, and set $T = 1$ in the main experiments. The step size $\alpha$ is randomly sampled from 1 to 100.

**Setups for Ablation Study.** We conduct extensive ablation studies on components, hyperparameters and routing strategies as shown in Section 4. For simplicity, we slightly modified the training setups. We train all methods using 2.5M TSP100 instances. The learning rate is decayed by 10 for the last 20% training instances. For the ablation on components (Fig. 3(a)), we set the attack steps as $T = 2$, and remove each component separately to demonstrate the effectiveness of each component in our proposed framework. For the ablation on hyperparameters (Fig. 3(b)), we train multiple models with $M \in \{2, 3, 4, 5\}$. For the ablation on routing strategies (Fig. 3(c)), we set $\mathcal{K} = B$ for M-Top$\mathcal{K}$, where $B = 64$ is the batch size, and $\mathcal{K} = 1$ for I-Top$\mathcal{K}$. The other training setups remain the same.

**Setups for Generalizability.** For the pretraining stage, we train MatNet (Kwon et al., 2021) on 5M ATSP20 instances following the original setups from Kwon et al. (2021). We further train a perturbation model by attacking it using reinforcement learning. Concretely, we use the dataset from Lu et al. (2023), consisting of 50 "tmat" class ATSP training instances that obey the triangle inequality, to train the perturbation model for 500 epochs. Adam optimizer is used with the learning rate of $1e - 3$. The maximum number of actions taken by the perturbation model is $T = 10$. After the pretraining stage, we use the pretrained model to initialize $M = 3$ models, and further adversarially train them. We fix the perturbation model and only update it using the latest model every $E = 10$ epochs (as discussed in Appendix B.3). After the $10_{th}$ epoch, there would be $M$ perturbation models corresponding to $M$ models. Following Lu et al. (2023), we use the fixed 1K clean instances for training. In the inner maximization, we generate $M$K local adversarial instances and 1K global adversarial instances using the perturbation models. However, since the perturbation model is not efficient (i.e., it needs to conduct the beam search to find the edges to be modified), we generate adversarial instances in advance and reuse them later. Then, in the outer minimization, we load all instances and forward them to each model using the jointly trained neural router. The

---

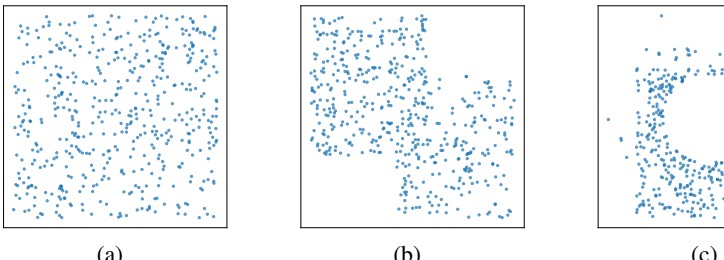

| (a) | (b) | (c) |

Figure 6: The generated TSP instances following the (a) Uniform distribution; (b) Rotation distribution; (c) Explosion distribution.

models are then adversarially trained for 20 epochs using the Adam optimizer with the learning rate of $4e - 5$, the weight decay of $1e - 6$ and the batch size of $B = 100$.

### D.2 GENERATION OF INSTANCES

We follow the instructions from Kool et al. (2018) and Zhou et al. (2023) to generate synthetic instances. Concretely, **1) Uniform Distribution:** The node coordinate of each node is uniformly sampled from the unit square $U(0, 1)$, as shown in Fig. 6(a). **2) Rotation Distribution:** Following Bossek et al. (2019), we mutate the nodes, which originally follow the uniform distribution, by rotating a subset of them (anchored in the origin of the Euclidean plane) as shown in Fig. 6(b). The coordinates of selected nodes are transformed by multiplying with a matrix $\begin{bmatrix} \cos(\varphi) & \sin(\varphi) \\ -\sin(\varphi) & \cos(\varphi) \end{bmatrix}$, where $\varphi \sim [0, 2\pi]$ is the rotation angle. **3) Explosion Distribution:** Following Bossek et al. (2019), we mutate the nodes, which originally follow the uniform distribution, by simulating a random explosion. Specifically, we first randomly select the center of explosion $v_c$ (i.e., the hole as shown in Fig. 6(c)). All nodes $v_i$ within the explosion radius $R = 0.3$ is moved away from the center of explosion with the form of $v_i = v_c + (R + s) \cdot \frac{v_c - v_i}{||v_c - v_i||}$, where $s \sim \text{Exp}(\lambda = 1/10)$ is a random value drawn from an exponential distribution.

In this paper, we mainly consider the distribution of node coordinates. For CVRP instances, the coordinate of the depot node $v_0$ is uniformly sampled from the unit square $U(0, 1)$. The demand of each node $\delta_i$ is randomly sampled from a discrete uniform distribution $\{1, \ldots, 9\}$. The capacity of each vehicle is set to $Q = \lceil 30 + \frac{n}{5} \rceil$, where $n \geq 50$ is the size of CVRP instances. The demand and capacity are further normalized to $\delta'_i = \delta/Q$ and 1, respectively.

**Set-XML100 Benchmark Instances.** The Set-XML100 (Queiroga et al., 2022) is a newly proposed benchmark dataset, including a broad range of distribution shifts, such as depot positioning ($A$), customer positioning ($B$), demand distribution ($C$), and average route size ($D$). Since it originally only contains instances with $n = 100$, we further generate instances with the sizes $n \in [125, 150, 175, 200]$ using its source code. Specifically, the four attributes are randomly sampled from the Cartesian product of $A\{1, 2, 3\} \times B\{1, 2, 3\} \times C\{1, 2, 3, 4, 5, 6, 7\} \times D\{1, 2, 3, 4, 5, 6\}$. The name of an instance follows the pattern of XML$\{n\}$_$\{ABCD\}$_$\{$id$\}$ in Table 7. Following the setups described in Appendix D.1, we run HGS (Vidal, 2022) to obtain their (sub-)optimal solutions.

### D.3 RESULTS ON BENCHMARK INSTANCES

We evaluate all methods (only trained on $n = 100$) on the real-world benchmark instances, including TSPLIB[5] (Reinelt, 1991) and CVRPLIB[6] (Set-X (Uchoa et al., 2017) and Set-XML100 (Queiroga et al., 2022)), where we choose representative instances with $n \in [100, 200]$. For Set-XML100, we randomly sample 5 instances with $n = 100$ from the original dataset, and generate extra 20 instances with $n \in [125, 150, 175, 200]$ following the data generation process. Besides, we also show results of Omni-VRP (Zhou et al., 2023), which is a recent work leveraging meta-learning techniques (Finn et al., 2017; Nichol et al., 2018) to deal with the OOD generalization of neural VRP methods. The results are obtained using their open-source models[7], which are trained on $n \in [50, 200]$ with diverse

---

[5] http://comopt.ifi.uni-heidelberg.de/software/TSPLIB95/tsp
[6] http://vrp.galgos.inf.puc-rio.br/index.php
[7] https://github.com/RoyalSkye/Omni-VRP

distributions. Moreover, we combine our method with the efficient active search (EAS) (Hottung et al., 2022) by running EAS-Lay and EAS-Emb on each instance following their setups (e.g., 1 run and 200 iterations), and report the best result. The results are shown in Tables. 5, 6 and 7.

Table 3: Performance evaluation against ROCO (Lu et al., 2023) over 1K ATSP instances.

| | Clean | | | | Fixed Adv. | | | |
| | (x1) Gap | Time | (x16) Gap | Time | (x1) Gap | Time | (x16) Gap | Time |
|---|---|---|---|---|---|---|---|---|
| LKH3 | 0.000% | 1s | 0.000% | 1s | 0.000% | 1s | 0.000% | 1s |
| Nearest Neighbour | 30.481% | – | 30.481% | – | 31.595% | – | 31.595% | – |
| Farthest Insertion | 3.373% | – | 3.373% | – | 3.967% | – | 3.967% | – |
| MatNet (1) | 0.784% | 0.5s | 0.056% | 5s | 0.931% | 0.5s | 0.053% | 5s |
| MatNet_AT (1) | 0.817% | 0.5s | 0.072% | 5s | 0.827% | 0.5s | 0.046% | 5s |
| MatNet_AT (3) | 0.299% | 1.5s | 0.028% | 15s | 0.319% | 1.5s | 0.023% | 15s |
| CNF (3) | **0.246%** | 1.5s | **0.022%** | 15s | **0.278%** | 1.5s | **0.015%** | 15s |

## D.4 RESULTS ON GENERALIZABILITY

We further evaluate the generalizability of CNF against other attacks. The results are shown in Table 3, where we evaluate all methods on 1K ATSP instances. For neural methods, we use the sampling with x1 and x16 instance augmentations following Kwon et al. (2021); Lu et al. (2023). The gaps are computed w.r.t. LKH3. From the results, we observe that 1) CNF is effective in mitigating the trade-off between the standard generalization and adversarial robustness; 2) together with the main results in Section 4, CNF is versatile to defend against various attacks for different neural methods.

## D.5 SENSITIVITY ANALYSES

In addition to the ablation study on the key hyperparameter (i.e., the number of models $M$), we further conduct sensitivity analysis on others, such as the optimizer $\in$ [Adam, SGD], batch size $\in$ [32, 64, 128], normalization layer $\in$ [batch, instance] and learning rate (LR) $\in [1e-3, 1e-4, 1e-5]$). The experiments are conducted on TSP100 following the setups of the ablation study as presented in Appendix D.1. The results are shown in Table 4, where we observe that the performance of our method could be further boosted by carefully tuning hyperparameters.

Table 4: Sensitivity Analyses on hyperparameters.

| Remark | Optimizer | Batch Size | Normalization | LR | Uniform (100) | Fixed Adv. (100) |
|---|---|---|---|---|---|---|
| Default | Adam | 64 | Instance | 1e-4 | 0.111% | 0.255% |
| | SGD | 64 | Instance | 1e-4 | 0.146% | 3.316% |
| | Adam | 32 | Instance | 1e-4 | 0.122% | 0.262% |
| | Adam | 128 | Instance | 1e-4 | **0.088%** | 0.311% |
| | Adam | 64 | Batch | 1e-4 | 0.114% | **0.247%** |
| | Adam | 64 | Instance | 1e-3 | 0.183% | 0.282% |
| | Adam | 64 | Instance | 1e-5 | 0.101% | 0.616% |

## D.6 ABLATION STUDY ON CVRP

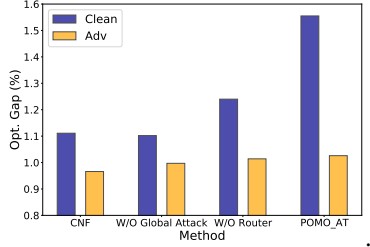

Figure 7: Ablation study on Components.

Similar to the ablation study on TSP (in Section 4), we further conduct the ablation study on CVRP. For simplicity, here we only consider investigating the role of each component in our method by removing them separately. The experiments are conducted on CVRP100, and the setups are kept the same as the ones presented in Appendix D.1. The results are shown in Fig. 7, which still verifies the effectiveness of the global attack and neural router in CNF.

# E    DISCUSSIONS

**Load Balancing.** In this paper, load balancing refers to each model being distributed with a similar or the same number of training instances from the instance set $\mathcal{X}$, in each outer minimization step. It could improve the training efficiency by evolving each model with similar quantities of data samples (based on the Cannikin Law), and avoid the dominant model or biased performance. The proposed neural router with the Top$\mathcal{K}$ operator could ensure such load balancing since each model is assigned exactly $\mathcal{K}$ instances based on the probability matrix predicted by the neural router.

**Larger Model.** In addition to training multiple models, increasing the number of parameters for a single model is also a way of enhancing the overall model capacity. However, technically, 1) a larger model needs more GPU memory, which puts more requirements on the single GPU device. It is also more complicated to enable parallel training on multiple GPUs compared with the multiple models; 2) currently, our method conducts AT upon the pretrained model, but there does not exist a larger pretrained model (e.g., larger POMO) in the literature. Despite the technical issues, we try to pre-train the larger POMO (e.g., 18 encoder layers with 3.64M parameters in total) on the uniform distributed data, and further conduct the vanilla AT. The performance is around 0.335% and 0.406% on clean and adversarial instances, respectively, which is inferior to the counterpart of multiple models (i.e., POMO_AT (3)). The superiority of multiple models may be attributed to its ensemble effect and the capacity in learning multiple diverse policies.

**Why using the best-performing model for the global attack?** The collaborative performance of our framework depends on the best-performing model $\theta_b$ (for each instance), since its solution will be chosen as the final solution during inference. The goal of inner maximization is to construct the adversarial instance that could successfully fool the framework. Intuitively, if we choose to attack other models (rather than $\theta_b$), the constructed adversarial instances may not successfully fool the best-performing model $\theta_b$, and therefore the final solution to the adversarial instance could still be good, which contradicts the goal. Therefore, to increase the success rate of attacking the framework and generate more diverse adversarial instances, for each clean instance, we choose the corresponding best-performing model $\theta_b$ as the global model in each iteration of the inner maximization.

**The variability of adversarial instances for the same initial model.** We take POMO (Kwon et al., 2020) as an example. During training, in each step of solution construction, the decoder of the neural solver selects the valid node to be visited by sampling from the probability distribution, rather than using argmax. Therefore, even though we initialize all models using the same pretrained model, given the same attack hyperparameters (e.g., attack iterations), the adversarial instances generated by each model are generally not the same at the beginning of the training.

**Generalizability to Other Domains.** Conceptually, the underlying idea of CNF could be applied to other domains with several adjustments. We take the image classification task as an example. Given the training data with its ground-truth label $(x, y)$, the global adversarial instance could be constructed by maximizing $\ell(x, y; \theta_b)$, where $\ell$ is the cross-entropy loss and $\theta_b = \arg\min_{\theta \in \Theta} \ell(x, y; \theta)$ is the best-performing model. In the outer minimization, we could also develop a neural router to distribute training instances to different models, with the objective of maximizing the improvement of ensemble performance. The cost matrix $R, R'$ in Eq. 6 could be filled with the probabilities on label $y$ predicted by each model. Despite the potential generalizability, the effectiveness of CNF in other domains needs to be empirically justified, and we hope CNF could inspire such future works.

**Advanced AT Techniques.** In this paper, we mainly focus on the vanilla AT (Madry et al., 2018). More advanced AT techniques, such as TRADES (Zhang et al., 2019), AT in RL (Pinto et al., 2017), ensemble-based adversarial training (Tramèr et al., 2018; Kariyappa & Qureshi, 2019; Pang et al., 2019; Yang et al., 2020a; 2021), and adversarial data augmentation (Xie et al., 2020; Cheng et al., 2020; Herrmann et al., 2022; Wen et al., 2022; Kong et al., 2022) could be considered as well. However, some of them may not be applicable to the discrete VRP domain due to their needs for ground-truth labels or the dependence on the imperceptible perturbation model. 1) TRADES is empirically effective for trading adversarial robustness off against standard generalization on the image classification task. Its loss function is formulated as $\mathcal{L} = \text{CE}(f(x), y) + \beta\text{KL}(f(x), f(\tilde{x}))$, where CE is cross-entropy, KL is the KL-divergence, $x$ is the clean instance, $\tilde{x}$ is the adversarial instance, $f(x)$ is the logit predicted by the model, and $y$ is the ground-truth label. By explicitly making the outputs of the network (logits) similar for $x$ and $\tilde{x}$, it could mitigate the trade-off. However, the above statement is conditional on the imperceptible perturbation model, where the ground-truth

labels of $x$ and $\tilde{x}$ are kept the same. As we discussed in Section 2.2, in the discrete VRPs, the perturbation model does not have such an imperceptible constraint. Therefore, it does not make sense to make the outputs of the model similar for $x$ and $\tilde{x}$, since the optimal solutions to $x$ and $\tilde{x}$ are not the same in the general case. 2) Another interesting direction is AT in RL, where the focus is the attack side rather than the defense side (e.g., most of the design in Pinto et al. (2017) focuses on the adversarial agent). Specifically, it jointly trains another agent (i.e., the attacker/adversary), whose objective is to impede the model (i.e., the first agent), to generate hard trajectories in a two-player zero-sum way. Its goal is to learn a policy that is robust to modeling errors in simulation or mismatch between training and test scenarios. While our work focuses on the defense side and aims to mitigate the trade-off between standard generalization and adversarial robustness. Moreover, this method is specific to RL while our framework has the potential to work with the supervised learning setting. Overall, it is non-trivial to directly apply this method to address our problem (e.g., the trade-off). But it is an interesting future research direction to design attack methods specific to RL (e.g., by training another adversarial agent or attacking each step of MDP). 3) Similar to the proposed CNF, ensemble-based adversarial training also leverages multiple models, but with a different motivation (e.g., reducing the adversarial transferability between models to defend against black-box adversarial attacks (Yang et al., 2020a)). Pang et al. (2019) needs the ground-truth labels to calculate the ensemble diversity. Yang et al. (2020a) depends on the misalignment of the distilled feature between the visual similarity and the classification result, and hence on the imperceptible perturbation model. Therefore, it is non-trivial to directly adapt them to the discrete VRP domain. Kariyappa & Qureshi (2019) proposes to decrease the gradient similarity loss to reduce the overall adversarial transferability between models, and Yang et al. (2021) further uses the model smoothness loss to improve the ensemble robustness. However, technically, their methods are computational expensive due to the needs to keep the computational graph before taking an optimization step. Compared with other baselines, their empirical results are not superior as well (see Section 4). 4) Adversarial data augmentation (Xie et al., 2020; Cheng et al., 2020; Herrmann et al., 2022; Wen et al., 2022; Kong et al., 2022) is found to be empirically effective in bolstering the generalization capabilities of deep learning models. But most works focus on the imperceptible perturbation model, where the adversarial perturbation is extremely small. In this paper, the global attack in CNF could also be viewed as one way of adversarial data augmentation, and we demonstrate the improved generalization of neural VRP solvers, even though the adversarial perturbation is relatively large.

**Practical Significance / Attack Scenarios.** Actually, there may not be a person or attacker to deliberately invade the VRP model in practice. However, the developers in a logistics enterprise should consider the adversarial robustness of neural VRP solvers as a sanity check before deploying them in the real world. The adversarial attack could be viewed as a way of measuring the worst-case performance of neural VRP solvers within the neighborhood of inputs. Without considering adversarial robustness, the neural solver may perform very poorly (see Fig. 1) when the testing instance pattern in the real world is 1) different from the training one, and 2) similar to the adversarial one. For example, considering an enterprise like Amazon, when some new (but not many) customers need to be added to the current configuration or instance, especially when their locations coincidentally lead to adversarial instances of the current solver (which corresponds to the node insertion attack presented in Appendix B.2), the model without considering adversarial robustness may output a very bad solution, and therefore resulting in unpleasant user experience and financial losses.

**Training Efficiency.** Following the training setups presented in Section 4, we show the training time of each method on TSP100 in Fig. 8. As mentioned in Section 5, the limitation of our proposed CNF is the increased training time complexity due to the need for synergistically training multiple models. Concretely, we empirically observe that CNF (3) takes longer time than the simple AT variants (e.g., POMO_AT (3), POMO_HAC (3)). However, we note that simply further training these methods cannot significantly increase their performance. For

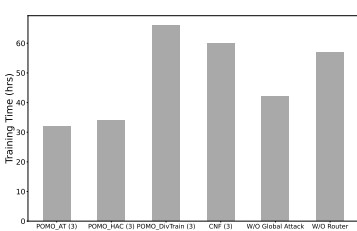

Figure 8: Training Time.

example, we try to train POMO_AT (3) for 60 hours, with the gap still inferior to ours (i.e., POMO_AT (3) vs. CNF (3): Uniform – 0.246% vs. 0.118%; Fixed Adv. – 0.278% vs. 0.236%). On the other hand, our training time is less than that of advanced AT training methods (e.g.,

POMO_DivTrain (3)). The above comparison indicates that our method can achieve a better trade-off to deliver better results within reasonable training time. Moreover, we find that the global attack generation consumes more training time than the neural router ($\sim$0.05 million parameters). During inference, the computational complexity only depends on the number of models, since the neural router is only activated during the training stage, and is discarded afterwards. Therefore, all methods with the same number of trained models have almost the same inference time.

**Attack Budget.** As we discussed in Section 2.2, we do not exert the imperceptible constraint on the perturbation model in VRPs. We further explain it from two perspectives. 1) Different from other domains, there is no theoretical guarantee to ensure the invariance of the optimal solution (or ground-truth label) to a clean VRP instance, given the imperceptible perturbation. A small change on even one node may induce a very different optimal solution. Therefore, we do not see the benefit of constraining the attack budget to a very small (i.e., imperceptible) range in VRP tasks. Moreover, even with the absence of imperceptible constraints on the adversary, unlike the graph learning setting, we do not observe a significant degradation on clean performance. It reveals that we don't need explicit imperceptible perturbation to restrain the changes of (clean) objective values. In our experiments, we set the attack budget within a reasonable range following the applied attack method (e.g., $\alpha \in [1, 100]$ for Zhang et al. (2022)). Our experimental results (in Section 4 and Appendix D) show that the proposed CNF is able to achieve a favorable balance, given different attack methods and their attack budgets. 2) In VRPs (or COPs), all generated adversarial instances are valid problem instances regardless of how much they differ from the clean instance. In this sense, the attack budget models the severity of a potential distribution shift between training data and test data (Geisler et al., 2022). This highlights the differences to other domains (e.g., computer vision), where unconstrained perturbations may lead to non-realistic or invalid data. Technically, the various attack budgets can help to generate diverse adversarial instances for training. Considering the above aspects, we believe our adversarial setting, including diverse but valid problem instances, may benefit the VRP community in developing a more general and robust neural solver. With that said, this paper could also be viewed as an attempt to improve the generalization of neural VRP solvers from the perspective of adversarial robustness.

**The Selection Basis of Attackers.** There are three attackers in the current literature of VRPs (or COPs) (Zhang et al., 2022; Geisler et al., 2022; Lu et al., 2023). We select the attacker based on its generality in VRPs. Specifically, 1) Geisler et al. (2022) is the early work that explicitly investigates the adversarial robustness in COPs. Their perturbation model needs to be sound and efficient, which means, given a clean instance and its optimal solution, the optimal solution to the adversarial instance can be directly derived without running a solver. However, this direct derivation requires the unique properties and theorems of certain problems (e.g., the intersection theorem (Cutler, 1980) in the Euclidean space for TSP), and hence is non-trivial to generalize to more complicated VRPs (e.g., CVRP). Moreover, their perturbation model is limited to attack the supervised neural solver (i.e., ConvTSP (Joshi et al., 2019)), since it needs to construct the adversarial instance by maximizing the loss function so that the model prediction is maximally different from the derived optimal solution. While in VRPs, reinforcement learning based methods (Kool et al., 2018; Kwon et al., 2020) are more appealing since they can gain even better performance without the need for optimal solutions. 2) Lu et al. (2023) requires that the optimal solution to the adversarial instance is no worse than that to the clean instance in theory, which may limit the problem space of adversarial instances. It focuses on the graph-based COPs (e.g., asymmetric TSP) and satisfies the requirement by lowering the cost of edges. Similar to Geisler et al. (2022), their method is not easy to design for VRPs with more constraints. Moreover, they resort to the black-box adversarial attack method by training a reinforcement learning based attacker, which may lead to a higher computational complexity and relatively low success rate of attacking. Therefore, for better generality, we choose Zhang et al. (2022) as the attacker in the main paper, which could be applied to different VRP variants and popular VRP solvers (Kool et al., 2018; Kwon et al., 2020). Moreover, we also evaluate the versatility of CNF against Lu et al. (2023) as presented in Appendix B.3 and D.4.

**Relationship between OOD Generalization and Adversarial Robustness.** Generally, adversarial robustness is one way to measure the generalization over the perturbed instance $\tilde{x}$ in the proximity of the clean instance $x$. In the context of VRPs (or COPs), adversarial instances are neither anomalous nor statistical defects since they correspond to valid problem instances regardless of how much they differ from $x$ (Geisler et al., 2022). In contrast to other domains, 1) the attack budget models the severity of a potential distribution shift between training data and test data; 2) the set of valid

problems is not just a low-dimensional manifold in a high-dimensional space, and hence the manifold hypothesis (Stutz et al., 2019) does not apply to combinatorial optimization. Therefore, it is critical for neural VRP solvers to perform well on adversarial instances when striving for a broader OOD generalization. Based on the experimental results, we empirically demonstrate that raising robustness against adversarial instances by CNF favorably promotes various types of generalization of neural VRP solvers (as shown in Section 4 and Appendix D.3), indicating the potential existence of neural VRP solvers with high generalization and robustness concurrently.

Table 5: Results on TSPLIB (Reinelt, 1991) instances. * Only for reference due to different problem settings, i.e., Omni_VRP is meta-trained on $n \in [50, 200]$, while others are adv-trained on $n = 100$.

| Instance | Opt. | POMO Obj. | POMO Gap | POMO_AT Obj. | POMO_AT Gap | POMO_HAC Obj. | POMO_HAC Gap | POMO_DivTrain Obj. | POMO_DivTrain Gap | Omni_VRP* Obj. | Omni_VRP* Gap | CNF Obj. | CNF Gap | CNF+EAS Obj. | CNF+EAS Gap |
|---|---|---|---|---|---|---|---|---|---|---|---|---|---|---|---|
| kroA100 | 21282 | 21420 | 0.65% | 21347 | 0.31% | 21308 | 0.12% | 21370 | 0.41% | **21305** | **0.11%** | 21308 | 0.12% | 21282 | 0.00% |
| kroB100 | 22141 | 22200 | 0.27% | 22211 | 0.32% | 22200 | 0.27% | **22199** | **0.26%** | 22650 | 2.30% | 22216 | 0.34% | 22199 | 0.26% |
| kroC100 | 20749 | 20799 | 0.24% | 20768 | 0.09% | **20753** | **0.02%** | 20768 | 0.09% | 20902 | 0.74% | 20758 | 0.04% | 20749 | 0.00% |
| kroD100 | 21294 | 21446 | 0.71% | 21391 | 0.46% | 21407 | 0.53% | 21435 | 0.66% | 21828 | 2.51% | **21353** | **0.28%** | 21294 | 0.00% |
| kroE100 | 22068 | 22259 | 0.87% | 22288 | 1.00% | 22167 | 0.45% | 22213 | 0.66% | 22239 | 0.77% | **22121** | **0.24%** | 22106 | 0.17% |
| eil101 | 629 | 630 | 0.16% | 630 | 0.16% | **629** | **0.00%** | 631 | 0.32% | 632 | 0.48% | 630 | 0.16% | 629 | 0.00% |
| lin105 | 14379 | 14477 | 0.68% | 14426 | 0.33% | 14408 | 0.20% | **14402** | **0.16%** | 14819 | 3.06% | 14403 | 0.17% | 14379 | 0.00% |
| pr107 | 44303 | 44678 | 0.85% | 47819 | 7.94% | **44596** | **0.66%** | 46285 | 4.47% | 44745 | 1.00% | 44719 | 0.94% | 44303 | 0.00% |
| pr124 | 59030 | 59389 | 0.61% | 59257 | 0.38% | 59385 | 0.60% | 59558 | 0.89% | 59238 | 0.35% | **59076** | **0.08%** | 59030 | 0.00% |
| bier127 | 118282 | 133042 | 12.48% | 118606 | 0.27% | 118608 | 0.28% | **118337** | **0.05%** | 121129 | 2.41% | 118841 | 0.47% | 118282 | 0.00% |
| ch130 | 6110 | 6119 | 0.15% | 6130 | 0.33% | 6115 | 0.08% | 6125 | 0.25% | 6251 | 2.31% | **6111** | **0.02%** | 6110 | 0.00% |
| pr136 | 96772 | 97983 | 1.25% | 100225 | 3.57% | 97617 | 0.87% | 100145 | 3.49% | 97780 | 1.04% | **97567** | **0.82%** | 97198 | 0.44% |
| pr144 | 58537 | 58935 | 0.68% | 59544 | 1.72% | 58913 | 0.64% | 59265 | 1.24% | 59571 | 1.77% | **58868** | **0.57%** | 58537 | 0.00% |
| ch150 | 6528 | 6554 | 0.40% | 6582 | 0.83% | 6556 | 0.43% | 6578 | 0.77% | 6586 | 0.89% | **6550** | **0.34%** | 6554 | 0.40% |
| kroA150 | 26524 | 26755 | 0.87% | 26898 | 1.41% | 26736 | 0.80% | 26813 | 1.09% | 26873 | 1.32% | **26722** | **0.75%** | 26524 | 0.00% |
| kroB150 | 26130 | 26405 | 1.05% | 26506 | 1.44% | **26379** | **0.95%** | 26467 | 1.29% | 26452 | 1.23% | 26494 | 1.39% | 26143 | 0.05% |
| pr152 | 73682 | **74249** | **0.77%** | 77537 | 5.23% | 75291 | 2.18% | 77127 | 4.68% | 74907 | 1.66% | 74876 | 1.62% | 73682 | 0.00% |
| rat195 | 2323 | 2486 | 7.02% | 2500 | 7.62% | 2461 | 5.94% | 2467 | 6.20% | **2417** | **4.05%** | 2449 | 5.42% | 2338 | 0.65% |
| kroA200 | 29368 | 29992 | 2.12% | 30222 | 2.91% | 29771 | 1.37% | 30143 | 2.64% | 29823 | 1.55% | **29755** | **1.32%** | 29435 | 0.23% |
| kroB200 | 29437 | 30298 | 2.92% | 30157 | 2.45% | 29890 | 1.54% | 30267 | 2.82% | **29814** | **1.28%** | 29862 | 1.44% | 29508 | 0.24% |

Table 6: Results on CVRPLIB (Set-X) (Uchoa et al., 2017) instances.

| Instance | Opt. | POMO Obj. | POMO Gap | POMO_AT Obj. | POMO_AT Gap | POMO_HAC Obj. | POMO_HAC Gap | POMO_DivTrain Obj. | POMO_DivTrain Gap | Omni_VRP* Obj. | Omni_VRP* Gap | CNF Obj. | CNF Gap | CNF+EAS Obj. | CNF+EAS Gap |
|---|---|---|---|---|---|---|---|---|---|---|---|---|---|---|---|
| X-n101-k25 | 27591 | 29282 | 6.13% | 29262 | 6.06% | 29315 | 6.25% | 29478 | 6.84% | 29442 | 6.71% | **28911** | **4.78%** | 27936 | 1.25% |
| X-n106-k14 | 26362 | 26961 | 2.27% | 26938 | 2.18% | 26906 | 2.06% | 26813 | 1.71% | 26990 | 2.38% | **26672** | **1.18%** | 26456 | 0.36% |
| X-n110-k13 | 14971 | 15154 | 1.22% | 15400 | 2.87% | 15215 | 1.63% | 15319 | 2.32% | 15285 | 2.10% | **15127** | **1.04%** | 14971 | 0.00% |
| X-n115-k10 | 12747 | 13877 | 8.86% | 13528 | 6.13% | 13409 | 5.19% | 13518 | 6.05% | **13240** | **3.87%** | 13928 | 9.26% | 13127 | 2.98% |
| X-n120-k6 | 13332 | 14574 | 9.32% | 15418 | 15.65% | 14907 | 11.81% | 14930 | 11.99% | 13944 | 4.59% | **13652** | **2.40%** | 13424 | 0.69% |
| X-n125-k30 | 55539 | 58412 | 5.17% | 58869 | 6.00% | **58116** | **4.64%** | 58571 | 5.46% | 58738 | 5.76% | 58238 | 4.86% | 56384 | 1.52% |
| X-n129-k18 | 28940 | 29565 | 2.16% | **29290** | **1.21%** | 29439 | 1.72% | 29411 | 1.63% | 29975 | 3.58% | 29348 | 1.41% | 29012 | 0.25% |
| X-n134-k13 | 10916 | 11315 | 3.66% | 11312 | 3.63% | 11343 | 3.91% | 11260 | 3.15% | 11302 | 3.54% | **11248** | **3.04%** | 11003 | 0.80% |
| X-n139-k10 | 13590 | 14084 | 3.64% | 14300 | 5.22% | 14011 | 3.10% | 14042 | 3.33% | 14019 | 3.16% | **13940** | **2.58%** | 13644 | 0.40% |
| X-n143-k7 | 15700 | 16382 | 4.34% | 16358 | 4.19% | 16190 | 3.12% | 16376 | 4.31% | 16602 | 5.75% | **15980** | **1.78%** | 15788 | 0.56% |
| X-n148-k46 | 43448 | 47613 | 9.59% | 47348 | 8.98% | 46751 | 7.60% | 47338 | 8.95% | 46438 | 6.88% | **45694** | **5.17%** | 44001 | 1.27% |
| X-n153-k22 | 21220 | 24354 | 14.77% | 23743 | 11.89% | 23785 | 12.09% | 23803 | 12.17% | **22810** | **7.49%** | 24643 | 16.13% | 22237 | 4.79% |
| X-n157-k13 | 16876 | 18294 | 8.40% | 17420 | 3.22% | 17503 | 3.72% | 17500 | 3.70% | **17107** | **1.37%** | 17640 | 4.53% | 17142 | 1.58% |
| X-n162-k11 | 14138 | 14986 | 6.00% | 15279 | 8.07% | 14975 | 5.92% | 15222 | 7.67% | **14595** | **3.23%** | 14794 | 4.64% | 14348 | 1.49% |
| X-n167-k10 | 20557 | **21294** | **3.59%** | 21435 | 4.27% | 21472 | 4.45% | 21584 | 5.00% | 21436 | 4.28% | 21658 | 5.36% | 20883 | 1.59% |
| X-n172-k51 | 45607 | 50351 | 10.40% | 49840 | 9.28% | 49190 | 7.86% | 50116 | 9.89% | **48399** | **6.12%** | 49926 | 9.47% | 46684 | 2.36% |
| X-n176-k26 | 47812 | 52889 | 10.62% | 51924 | 8.60% | 52541 | 9.89% | 52261 | 9.31% | **51332** | **7.36%** | 53420 | 11.73% | 49827 | 4.21% |
| X-n181-k23 | 25569 | 26969 | 5.48% | 26915 | 5.26% | 26867 | 5.08% | 26924 | 5.30% | **26088** | **2.03%** | 26213 | 2.52% | 25855 | 1.12% |
| X-n186-k15 | 24145 | 25734 | 6.58% | 25659 | 6.27% | 25620 | 6.11% | 25635 | 6.17% | **24768** | **2.58%** | 26109 | 8.13% | 24635 | 2.03% |
| X-n190-k8 | 16980 | 18064 | 6.38% | 17690 | 4.18% | 17824 | 4.97% | **17594** | **3.62%** | 17645 | 3.92% | 17770 | 4.65% | 17416 | 2.57% |
| X-n195-k51 | 44225 | 50296 | 13.73% | 49923 | 12.88% | 50015 | 13.09% | 49188 | 11.22% | **47477** | **7.35%** | 48823 | 10.40% | 45417 | 2.70% |
| X-n200-k36 | 58578 | 62094 | 6.00% | 62486 | 6.67% | 62243 | 6.26% | 62720 | 7.07% | 61496 | 4.98% | **61260** | **4.58%** | 60080 | 2.56% |

Table 7: Results on CVRPLIB (Set-XML100) (Queiroga et al., 2022) instances.

| Instance | (Sub-)Opt. | POMO | | POMO_AT | | POMO_HAC | | POMO_DivTrain | | Omni-VRP* | | CNF | | CNF+EAS | |
|---|---|---|---|---|---|---|---|---|---|---|---|---|---|---|---|
| | | Obj. | Gap | Obj. | Gap | Obj. | Gap | Obj. | Gap | Obj. | Gap | Obj. | Gap | Obj. | Gap |
| XML100_1113_01 | 14740 | 15158 | 2.84% | **15048** | **2.09%** | 15057 | 2.15% | 15145 | 2.75% | 15076 | 2.28% | 15079 | 2.30% | 14763 | 0.16% |
| XML100_1121_01 | 25764 | 27686 | 7.46% | 27565 | 6.99% | 27520 | 6.82% | 27765 | 7.77% | 27308 | 5.99% | **27134** | **5.32%** | 25830 | 0.26% |
| XML100_2151_01 | 26132 | 27777 | 6.29% | 27637 | 5.76% | 27675 | 5.90% | 27752 | 6.20% | 27753 | 6.20% | **26943** | **3.10%** | 26218 | 0.33% |
| XML100_3223_01 | 15031 | 15316 | 1.90% | 15198 | 1.11% | 15178 | 0.98% | 15182 | 1.00% | 15288 | 1.71% | **15171** | **0.93%** | 15057 | 0.17% |
| XML100_3332_01 | 28804 | 29509 | 2.45% | 29374 | 1.98% | 29358 | 1.92% | 29666 | 2.99% | 29625 | 2.85% | **29234** | **1.49%** | 28948 | 0.50% |
| XML125_2344_01 | 12330 | **12519** | **1.53%** | 12745 | 3.37% | 12582 | 2.04% | 12755 | 3.45% | 12950 | 5.03% | 12671 | 2.77% | 12350 | 0.16% |
| XML125_2353_01 | 14786 | 15212 | 2.88% | 15317 | 3.59% | 15229 | 3.00% | 15310 | 3.54% | 15426 | 4.33% | **15172** | **2.61%** | 14882 | 0.65% |
| XML125_3171_01 | 60336 | 64774 | 7.36% | 63092 | 4.57% | 63845 | 5.82% | 63171 | 4.70% | **62911** | **4.27%** | 64955 | 7.66% | 61424 | 1.80% |
| XML125_3276_01 | 7823 | 9038 | 15.53% | 8674 | 10.88% | 8507 | 8.74% | 8627 | 10.28% | 8822 | 12.77% | **8364** | **6.92%** | 7912 | 1.14% |
| XML125_3353_01 | 23244 | 24090 | 3.64% | 23839 | 2.56% | **23644** | **1.72%** | 23963 | 3.09% | 24098 | 3.67% | 23977 | 3.15% | 23289 | 0.19% |
| XML150_1126_01 | 11019 | 12437 | 12.87% | 12430 | 12.81% | 12244 | 11.12% | 11989 | 8.80% | 12462 | 13.10% | **11967** | **8.60%** | 11147 | 1.16% |
| XML150_1131_01 | 43844 | 46889 | 6.95% | 47601 | 8.57% | 46748 | 6.62% | 47572 | 8.50% | 46664 | 6.43% | **45556** | **3.90%** | 44250 | 0.93% |
| XML150_1144_01 | 17397 | 17909 | 2.94% | 17941 | 3.13% | **17852** | **2.62%** | 17878 | 2.76% | 18127 | 4.20% | 18004 | 3.49% | 17486 | 0.51% |
| XML150_3161_01 | 64484 | 68402 | 6.08% | 68833 | 6.74% | 68112 | 5.63% | 68970 | 6.96% | 67884 | 5.27% | **67673** | **4.95%** | 65051 | 0.88% |
| XML150_3251_01 | 50158 | 53902 | 7.46% | 54436 | 8.53% | 53317 | 6.30% | 54583 | 8.82% | 53469 | 6.60% | **52867** | **5.40%** | 50809 | 1.30% |
| XML175_1152_01 | 34440 | 36764 | 6.75% | 37295 | 8.29% | 36903 | 7.15% | 36897 | 7.13% | 36240 | 5.23% | **35630** | **3.46%** | 34764 | 0.94% |
| XML175_1246_01 | 9813 | 10843 | 10.50% | 11862 | 20.88% | **10760** | **9.65%** | 11133 | 13.45% | 10847 | 10.54% | 10875 | 10.82% | 10244 | 4.39% |
| XML175_1311_01 | 44406 | 49254 | 10.92% | 47595 | 7.18% | 47210 | 6.31% | 48042 | 8.19% | 46049 | 3.70% | **45616** | **2.72%** | 44459 | 0.12% |
| XML175_1344_01 | 17087 | 17924 | 4.90% | 17998 | 5.33% | **17864** | **4.55%** | 18136 | 6.14% | 17915 | 4.85% | 18097 | 5.91% | 17333 | 1.44% |
| XML175_3366_01 | 13950 | 15587 | 11.73% | 16802 | 20.44% | 15829 | 13.47% | 16356 | 17.25% | 15295 | 9.64% | **15117** | **8.37%** | 14417 | 3.35% |
| XML200_1215_01 | 16457 | 23513 | 42.88% | 20732 | 25.98% | 20265 | 23.14% | 19764 | 20.09% | **17174** | **4.36%** | 17736 | 7.77% | 17006 | 3.34% |
| XML200_1362_01 | 35840 | 38333 | 6.96% | 38246 | 6.71% | 38310 | 6.89% | 38558 | 7.58% | **37119** | **3.57%** | 37331 | 4.16% | 36221 | 1.06% |
| XML200_3133_01 | 38372 | 40344 | 5.14% | 40352 | 5.16% | 40372 | 5.21% | 40194 | 4.75% | **39231** | **2.24%** | 40505 | 5.56% | 38984 | 1.59% |
| XML200_3134_01 | 29402 | 31292 | 6.43% | 31200 | 6.12% | 31308 | 6.48% | 30966 | 5.32% | **30694** | **4.39%** | 31693 | 7.79% | 30006 | 2.05% |
| XML200_3315_01 | 22508 | 25341 | 12.59% | 28500 | 26.62% | 26775 | 18.96% | 27046 | 20.16% | **23432** | **4.11%** | 24212 | 7.57% | 23109 | 2.67% |

