# OpenReview forum: "Collaboration! Towards Robust Neural Methods for Vehicle Routing Problems"
_ICLR.cc/2024/Conference — Submitted to ICLR 2024_

### Official Review · Reviewer_4zRf · 2023-10-31

**Soundness:** 3 good
**Presentation:** 4 excellent
**Contribution:** 2 fair
**Rating:** 6
**Confidence:** 4

**Summary:**

This paper presents a systematic study of the adversarial training of neural network solvers for TSP and CVRP. The methodology part extends the adversarial training pipeline by ensembling multiple models, leading to the so-called CNF approach in this paper. Experiments are conducted on TSP and CVRP, with adversarial samples mainly generated by (Zhang et al., 2022).

**Strengths:**

* This paper is well-written and easy to follow. I could see the efforts of the authors in the organization and the figures.
* Robustness in solving routing problems is an important measure and worth studying.
* The experiment study seems extensive and in general seems sound.

**Weaknesses:**

* The authors believe that model degeneration on clean data is an important issue, and propose the Collaborative Neural Framework to solve it. However, the motivation for developing such a collaborative framework is not clear. The authors only mentioned that collaboration will mitigate existing issues in adversarial training, but do not explain why. From my side, the reason for the improvements brought by "collaboration" is quite straightforward and trivial: "collaboration" in this paper means training multiple models in adversarial training and using all of them to predict on the same problem instance. Since we are studying an optimization problem where solution evaluation is very fast, we can easily generate multiple predictions using multiple models and pick the best one. Therefore, it is not surprising that "collaborated" models are way better than standalone AT models.
* One step further on the previous point, it is not surprising to see that the performance of Collaborative Neural Framework is kept on clean instances after adversarial training. One can achieve similar results in Figure 1 by simply freezing one model on the clean data and doing adversarial training on the other models. The "collaborated" model will never degenerate on the clean data.
* The improvements brought by the interesting neural router module seem only marginal. As read from Table 1, there are significant performance improvements when 1) from POMO to POMO_AT (add adversarial training), and 2) from POMO_AT (1) to POMO_AT (3) (use 3 models together, i.e., collaboration). Adversarial training seems not an original contribution in this paper; Collaboration, as discussed above, is not surprising to bring a significant improvement. While with the only technically sound module, the improvement from POMO_AT (3) to CNF (3) is not that significant.
* The authors made the claim on Page 2 that "simply increasing the model capacity" will not help Adversarial Training to generalize better, but I do not find any experimental evidence to support that in the main paper.
### Minor Points
* In the title: in combinatorial optimization's convention, VRP is usually considered different from TSP. However, in this paper, the authors seem to call both TSP and CVRP "subsets of VRP". I believe it will cause confusion and would suggest changing the naming to "routing problems".
* Please explain "OOD" in Section 4.3 to make it self-contained.
* Not sure if it is proper to call all Reinforcement Learning methods as "REINFORCE" in Eq (1) because the RL algorithms deployed nowadays usually integrate many more tricks than the vanilla form in Eq (1).

**Questions:**

* The authors mentioned three papers on the adversarial robustness of CO, and based the major experiment study in the main paper on (Zhang et al. 2022). From my understanding, the other two papers (Geisler et al., 2022) and (Lu et al., 2023) considered the "hardness" of the CO problem itself (i.e., there are some guarantees on the optimal objective), while (Zhang et al., 2022) only considered the objective score solved by an existing solver. Can the author justify the reason of selecting (Zhang et al., 2022) as the main experiment protocol?
* Do the neural network know the behavior of the attacker during training? I.e., are the adversarial data points generated by the attacker in the test dataset?

---

> ### Author Response · Authors · 2023-11-17
> **Response to Reviewer 4zRf: Part 1**
>
> We sincerely appreciate the reviewer for taking your time and efforts in reviewing our paper and providing constructive feedback. We present detailed responses below, where W and Q refer to the weakness and question, respectively.
>
> **W1: What is the motivation to develop CNF and why does CNF work? It is not surprising that "collaborated" models are way better than standalone AT models.**
>
> **Motivation of CNF.** (1) The study on robustness in combinatorial optimization problems (COPs) is critical. It is highly related to the practicality and reliability of neural solvers. Recent works start paying more attention to robustness but most of them only investigate the attack methods. How to defend against adversarial instances is underexplored. In this paper, we focus on not only effectively defending against various attacks but also largely improving the performance on clean instances. Such a study may benefit the development of versatile neural solvers in different real-world scenarios. (2) One intuitive motivation to solve the above issue is “collaboration”. While we agree with the reviewer that the collaboration is straightforward, it is commonly used and shown effectiveness in different domains. However, according to our experiments, simple training multiple neural solvers cannot achieve a favorable trade-off between clean and adversarial performance (see Fig. 1 in Section 1). Therefore, it further motivates us to develop CNF, which is a more elegant collaborative framework to achieve a better trade-off. We would like to note that our CNF is the first work to apply such a collaborative paradigm to the robustness study in the VRP (or COP) domain.
>
> **Source of CNF power.** Instead of simply training multiple models, the effectiveness of the proposed collaboration mechanism in CNF can be attributed to its diverse adversarial data generation and the reasonable exploit of overall model capacity. As shown in the ablation study (Fig. 3(a)), the diverse adversarial data generation is helpful in further improving the adversarial robustness (see results of `CNF vs. W/O Global Attack`). Meanwhile, the neural router has a bigger effect in mitigating the trade-off (see results of `CNF vs. W/O Router`). Intuitively, by distributing instances to suitable submodels for training, each submodel might be stimulated to have its own expert area. Accordingly, the overlap of their vulnerability areas may be decreased, which could promote the collaborative performance of CNF. As shown in Fig. 5 (in the updated manuscript), not all models perform well on each kind of instance. Such diversity in submodels contributes to the mitigation of the trade-off between standard generalization and adversarial robustness, thereby significantly outperforming vanilla AT with multiple models.
>
> **Significance of our results.** CNF not only outperforms the standalone AT models, but also outperforms the collaborated version of vanilla AT, DivTrain [1], and HAC [2] by a large margin as shown in Table 1. These baselines are fairly competitive. Specifically, we would like to note that: (1) DivTrain, which belongs to the diversity-enhanced ensemble training method, also includes a collaboration among different models to enhance the sample diversity, and (2) all the baseline methods also generate multiple solutions using multiple models and pick the best one. Moreover, we also compare with another ensemble adversarial training method (i.e., TRS [3]) as suggested by the other reviewer, and the results still manifest the superiority of our method. We kindly refer the detailed results to our response to [`Reviewer ko2G Q2`](https://openreview.net/forum?id=zEOnlJaRKp&noteId=QN3T9vEUDz). In summary, the CNF gains significantly better comprehensive performance than various baselines, rather than trivial improvement over standalone AT models.
>
> [1] Kariyappa, Sanjay, and Moinuddin K. Qureshi. "Improving adversarial robustness of ensembles with diversity training." *arXiv preprint arXiv:1901.09981* (2019).
> [2] Zhang, Zeyang, et al. "Learning to solve traveling salesman problem with hardness-adaptive curriculum." In AAAI 2022.
> [3] Yang, Zhuolin, et al. "TRS: Transferability reduced ensemble via promoting gradient diversity and model smoothness." In NeurIPS 2021.

---

> ### Author Response · Authors · 2023-11-17
> **Response to Reviewer 4zRf: Part 2**
>
> **W2: How about simply freezing one model on the clean data and doing adversarial training on the other models?**
>
> Thanks for your interesting question. Yes. We could freeze one model and conduct AT on the other two models. In this case, however, the training of other models (without collaboration) is degenerated to the vanilla AT (i.e., POMO_AT (1)). Following the reviewer’s suggestion, we conducted an extra experiment. We observe that this kind of method achieves 0.144% (Uniform) and 0.329% (Fixed Adv.) gaps on TSP100, while CNF can achieve much better results, i.e., 0.118\% (Uniform) and 0.236\% (Fixed Adv.). The issue is that the clean performance of this kind of method is upper-bounded by that of the frozen model, while the proposed CNF has the potential to improve the clean and adversarial performance concurrently.
>
> From another point of view, the method mentioned by the reviewer could be viewed as replacing the neural router with a simple and explicit heuristic in our framework. It is equivalent to the CNF application, where one model is trained on clean instances and the others are trained on adversarial instances. In contrast, the proposed neural router in CNF could learn a better instance routing (or gating) policy, with the objective towards maximizing the overall collaboration performance.
>
> **W3: The improvements brought by the interesting neural router module seem only marginal.**
>
> As we discussed in the response to `W1`, the neural router plays an important role in mitigating the trade-off and improving the collaboration performance, due to its ability to reasonably exploit the overall model capacity. We would like to note that the trade-off is an important research topic in the literature of adversarial ML. Tremendous research [4-6] has been conducted only to mitigate this issue (since it precludes the practicality of adversarial training). However, such research is still a void in the COP domain. Our empirical results demonstrate that, within a reasonable computational budget, existing defensive methods (e.g., vanilla AT, HAC, and DivTrain) cannot solve the trade-off in VRPs, while the proposed CNF could achieve much better results, indicating the existence of neural VRP solvers with high generalization and robustness simultaneously [7].
>
> Moreover, by enlarging 3 times the model capacity, the performance improvement from `POMO_AT (1) `to `POMO_AT (3)` is around 0.11% and 0.27% on Uniform TSP100 and TSP200, respectively. While with almost the same number of parameters, the performance improvement from `POMO_AT (3)` to `CNF (3)` achieves 0.14% and 1.27%. Therefore, we reckon the improvement from POMO_AT (3) to CNF (3) is significant.
>
> Last but not least, we have conducted the ablation study to verify the effectiveness of the neural router. Specifically, as shown in Fig. 3(a), by comparing `CNF vs. W/O Router`, we observe that the neural router can greatly improve the standard generalization on the clean data, and hence plays an important role in mitigating the trade-off.
>
> [4] Tsipras, Dimitris, et al. "Robustness may be at odds with accuracy." In ICLR 2019.
> [5] Zhang, Hongyang, et al. "Theoretically principled trade-off between robustness and accuracy." In ICML 2019.
> [6] Yang, Yao-Yuan, et al. "A closer look at accuracy vs. robustness." In NeurIPS 2020.
> [7] Geisler, Simon, et al. "Generalization of neural combinatorial solvers through the lens of adversarial robustness." In ICLR 2022.

---

> ### Author Response · Authors · 2023-11-17
> **Response to Reviewer 4zRf: Part 3**
>
> **W4: Experimental evidence of the effect of simply increasing the model capacity.**
>
> Thanks for pointing this out. In fact, we have provided the empirical evidence in Fig. 1 with the analyses in Section 1 of the original paper. As mentioned there, we would like to justify that simply increasing the model capacity can partially mitigate the trade-off, but it is not a computationally efficient way to solve the problem. Instead, our goal in this paper is to mitigate the trade-off within a reasonable computational budget. From the empirical results in Fig. 1, we observe that CNF with 3 models has already achieved decent performance, which even outperforms the vanilla AT trained with more models, demonstrating a better trade-off between performance and computational budget.
>
> To further clarify the reviewer’s question, we additionally increase the number of trained models to demonstrate the superiority of the proposed CNF on TSP100. The results are shown below, where the number in the bracket denotes the number of trained models. We observe that the proposed CNF achieves much better clean performance even compared with vanilla AT with 9 models. We have included the new results in Fig. 1.
>
> | Method       | Num. of Parameters |  Uniform   | Fixed Adv. |
> | ------------ | :----------------: | :--------: | :--------: |
> | POMO\_AT (3) |       3.81 M       |   0.255%   |   0.295%   |
> | POMO\_AT (5) |       6.35 M       |   0.215%   |   0.257%   |
> | POMO\_AT (7) |       8.89 M       |   0.187%   |   0.242%   |
> | POMO\_AT (9) |      11.43 M       |   0.177%   |   0.231%   |
> | CNF (3)      |       3.96 M       |   0.118%   |   0.236%   |
> | CNF (5)      |       6.60 M       | **0.066%** | **0.223%** |
>
> Besides increasing the number of models, we also try to increase the number of parameters in a single model (i.e., a larger model). The results and discussion can be found in Appendix E.
>
> **W5: Changing the title naming to "routing problems".**
>
> Thanks for your suggestion. We understand the reviewer’s viewpoint, and have changed the title to *"Collaboration! Towards Robust Neural Methods for Routing Problems"* in the updated version.
>
> **W6: Please explain "OOD" in Section 4.3 to make it self-contained.**
>
> In Section 4.3, OOD generalization refers to the out-of-distribution generalization, which is a crucial challenge faced by the current neural VRP solvers. In the original paper, we ignore the explanation, considering the fact that it is a commonly used concept in deep learning. However, we understand the reviewer’s viewpoint that all concepts in the paper should be self-contained. Hence we have added the explanation in the updated manuscript.
>
> **W7: Not sure if it is proper to call all Reinforcement Learning methods as "REINFORCE".**
>
> The "REINFORCE" refers to the specific algorithm proposed by [8], which is widely used in popular neural construction-based VRP solvers with strong performance [9-12]. We also agree that developing more advanced RL algorithms to solve VRPs is a crucial and interesting research direction in the future.
>
> [8] Williams, Ronald J. "Simple statistical gradient-following algorithms for connectionist reinforcement learning." *Machine learning* 8 (1992): 229-256.
> [9] Kool, Wouter, Herke van Hoof, and Max Welling. "Attention, Learn to Solve Routing Problems!." In ICLR 2019.
> [10] Kwon, Yeong-Dae, et al. "Pomo: Policy optimization with multiple optima for reinforcement learning." In NeurIPS 2020.
> [11] Kim, Minsu, Junyoung Park, and Jinkyoo Park. "Sym-nco: Leveraging symmetricity for neural combinatorial optimization." In NeurIPS 2022.
> [12] Chen, Jinbiao, et al. "Efficient Meta Neural Heuristic for Multi-Objective Combinatorial Optimization." In NeurIPS 2023.

---

> ### Author Response · Authors · 2023-11-17
> **Response to Reviewer 4zRf: Part 4**
>
> **Q1: Can the author justify the reason for selecting (Zhang et al., 2022) as the main experiment protocol?**
>
> Thanks for your question. As we discussed in the related work section (Appendix A), there exist three typical attackers in VRPs [2, 7, 13]. We select the attacker according to their generality in VRPs:
>
> * [7] is an early work investigating the adversarial robustness in COPs, in which the authors claim that the perturbation model needs to be sound and efficient. It means that, given a clean instance and its optimal solution, the optimal solution to the adversarial instance should be directly derived without running a solver. However, this direct derivation requires unique properties and theorems of certain problems (e.g., the intersection theorem [14] in the Euclidean space for TSP). Hence the attacker in [7] is non-trivial to generalize to more complicated VRPs (e.g., CVRP). Moreover, its perturbation model is limited to attack the supervised neural solver (i.e., ConvTSP [15]), since it needs to construct the adversarial instance by maximizing the loss function so that the model prediction is maximally different from the derived optimal solution. But in VRPs, DRL is more appealing, which can yield better neural solvers than supervised learning without the need for optimal solutions.
> * The work in [13] theoretically requires that the optimal solution to the adversarial instance should not be worse than that to the clean instance, which may limit the problem space of adversarial instances. It is also limited to the graph-based COPs (i.e., asymmetric TSP in [13]) to satisfy the requirement by lowering the cost of edges. Similar to [7], their method is not easy to design for VRPs with more constraints. Moreover, it resorts to the black-box adversarial attack method by training a reinforcement learning based attacker, which may lead to higher computational complexity and relatively low success rate of attacking.
>
> Considering the above facts, we choose [2] in the main experiment for better generality. Specifically, it is applicable to different problems (e.g., TSP and CVRP) and VRP solvers [9, 10]. Please note that except for the attacker in [2], we also apply our method to defend against the attacker in [13] (please see Appendix B.3 and D.4.), which further verifies the versatility of the proposed CNF. In summary, our CNF can successfully defend against various attacks, indicating its favorable generality and versatility.
>
> [2] Zhang, Zeyang, et al. "Learning to solve traveling salesman problem with hardness-adaptive curriculum." In AAAI 2022.
> [7] Geisler, Simon, et al. "Generalization of neural combinatorial solvers through the lens of adversarial robustness." In ICLR 2022.
> [9] Kool, Wouter, Herke van Hoof, and Max Welling. "Attention, Learn to Solve Routing Problems!." In ICLR 2019.
> [10] Kwon, Yeong-Dae, et al. "Pomo: Policy optimization with multiple optima for reinforcement learning." In NeurIPS 2020.
> [13] Lu, Han, et al. "ROCO: A General Framework for Evaluating Robustness of Combinatorial Optimization Solvers on Graphs." In ICLR 2023.
> [14] Cutler, M. "Efficient special case algorithms for the n‐line planar traveling salesman problem." *Networks* 10.3 (1980): 183-195.
> [15] Joshi, Chaitanya K., Thomas Laurent, and Xavier Bresson. "An efficient graph convolutional network technique for the travelling salesman problem." *arXiv preprint arXiv:1906.01227*(2019).
>
> **Q2: Does the neural network know the behavior of the attacker during training?**
>
> The adversarial data generated by the current models (in the inner maximization) is used for training in the outer minimization. In this sense, the neural network knows the behavior of the attacker during training.
>
> The adversarial test datasets are generated by the pretrained POMO and the final checkpoint of each method, corresponding to the evaluation on `Fixed Adv.` and `Adv.` metrics, where the `Fixed Adv.` and `Adv.` are the black-box and conventional white-box (e.g., the attacker knows everything about the models) metrics commonly used in adversarial ML.

---

> ### Author Response · Authors · 2023-11-23
>
> Dear Reviewer 4zRf:
>
> Please kindly let us know if you still have any concerns given our response. In light of the tight deadline (today) for the author-reviewer discussion, it would be highly appreciated if you could pose them early (if any). We thank you again for your valuable comments and suggestions.

---

> ### Comment · Reviewer_4zRf · 2023-11-23
>
> Thank you so much for the detailed feedback. Generally, they seem sound to me, and for "W7" I believe the papers you mentioned use modern _Reinforcement Learning_, not the simple and classic REINFORCE method to estimate the gradient.
>
> I raise my score to 6 and please make sure to include the new results and discussions in future versions.

---

> > ### Author Response · Authors · 2023-11-23
> >
> > We sincerely appreciate the reviewer for acknowledging our rebuttal and raising the score. We promise to include the new results and discussions in future versions. Thanks again!

---

### Official Review · Reviewer_APh6 · 2023-11-01

**Soundness:** 2 fair
**Presentation:** 3 good
**Contribution:** 2 fair
**Rating:** 5
**Confidence:** 3

**Summary:**

To solve vehicle routing problems’ (VRPs) vulnerability to adversarial examples and existing adversarial training methods do not strike a good balance between generalization (on clean instances) and robustness (on adversarial instances). The Collaborative Neural Framework (CNF) is proposed by the authors, but some descriptions in the paper are not very accurate and clear. The novelty is also limited. The following are some of my doubts and suggestions for the paper, hoping to improve the quality of the paper.

**Strengths:**

The content of this paper is practical and valuable. The experiment of the paper is relatively sufficient.

**Weaknesses:**

(1) This paper describes adversarial examples in the field of image processing, but does not give a formal definition of adversarial examples in the field of VRPS, and suggests adding a formal definition of adversarial examples in the field of VRPS.

(2) When the number of trained models is listed in Figure 1, it is only increased to 5. Perhaps when the number of models increases, the traditional adversarial training method can exceed the method proposed in the paper. Therefore, it is suggested to increase the number of trained models to demonstrate the effectiveness of the method.

(3) The basis for selecting an attacker is not elaborated, and it is suggested to prove the generality of the selected attacker.

(4) At the end of the paper, experiments are conducted on the out of distribution data, but the relationship between OOD data’s performance and the adversarial robustness is not explained in detail, so it is suggested to elaborate.

**Questions:**

(1) During the Outer Minimization of CNF, there are three instance types, namely "ori", "local attack" and "global attack". How to prove that the instance of "global attack" improves the effectiveness of the method? It is suggested to add experiments in this part or give theoretical explanations.
(2) In the Outer Minimization stage of CNF, a neural router is trained. Will training a neural router seriously increase the training time? It is suggested to clarify in the paper.

---

> ### Author Response · Authors · 2023-11-17
> **Response to Reviewer APh6: Part 1**
>
> We greatly thank the reviewer for reviewing our paper and providing constructive comments. We list detailed responses below, where we adopt W and Q to mark responses to weaknesses and questions, respectively.
>
> **W1: A formal definition of adversarial examples in the field of VRPs.**
>
> Thanks for your suggestion. This paper aims to propose a framework to defend against different attack approaches, each of which has their own way to generate adversarial examples (please see Appendix B). Here, we try to give a unified expression of the adversarial examples in the VRP field, as below.
>
> We define that adversarial instance in VRP as the instance that is obtained by the perturbation model $G$ within the neighborhood of the clean instance, and is underperformed by the current model or solver. Formally, given a clean VRP instance $x=\{x_c, x_d\}$, where $x_c \in \mathcal{N}_c$ is the continuous attribute (e.g., node coordinates) within the valid range $\mathcal{N} _c$, and $x_d \in \mathcal{N} _d$ is the discrete attribute (e.g., node demand) within the valid range $\mathcal{N} _d$, the adversarial instance $\tilde{x}$ is found by the perturbation model $G$ around the clean instance $x$, on which the current model may be vulnerable. Specifically, the adversarial instance $\tilde{x}$ is constructed by adding crafted perturbations $\gamma _{x_c}, \gamma _{x_d}$ to the corresponding attribute of the clean instance, and then project them back to the valid domain: $\tilde{x} _c = \Pi _{\mathcal{N} _c}(x_c + \alpha \cdot \gamma _{x_c}),\ \tilde{x} _d = \Pi _{\mathcal{N} _d}(x_d + \alpha \cdot \gamma _{x_d})$, where $\Pi$ denotes the projection operator (e.g., min-max normalization), and $\alpha$ denotes the attack budget. The crafted perturbations $\gamma$ could be obtained by various perturbation models, such as the one in Eq. (7), e.g., $\gamma _c = \nabla _{{x}_c} \ell (x; \theta), \ \gamma _d=\nabla _{{x}_d} \ell (x; \theta)$, where $\theta$ and $\ell$ denote the model parameters and the loss function, respectively. We omit the attack step $t$ for notation simplicity.
>
> The above definition of adversarial examples is generally applicable to different attacks in VRPs. In our experiments, we evaluate against two perturbation models (in Section 4 and Appendix D.4). We add the above definition, and refer details of each perturbation model (i.e., attacker) to Appendix B. Moreover, we also provide an illustration of the adversarial instance in Fig. 4.
>
> **W2: Increase the number of trained models.**
>
> Yes, with the number of models increasing, the traditional adversarial training (AT) method can further improve the performance. But the proposed CNF can improve the performance as well (as shown in Fig. 3 (b)). We would like to note that the proposed CNF with 3 models has already outperformed the traditional AT with 7 models (~1M parameters per model), showing a better trade-off between performance and computational budget. On the other hand, the motivation of this paper is “*how to effectively synergize multiple models to achieve favorable overall performance on both clean and adversarial instances within a reasonable computational budget”*. To this end, we keep the same number of trained models for both vanilla AT and CNF. Based on our empirical results, we observe that the CNF outperforms the traditional AT by a large margin, demonstrating a better synergy of models within the proposed CNF.
>
> Following the reviewer’s suggestion, we further increase the number of trained models to demonstrate the superiority of our method on TSP100. The results are shown in the below table, where we record the number of trained models in the bracket. As observed, our method consistently outperforms the traditional AT, even if equipped with more models. We have updated the new empirical results in Fig. 1.
>
> | Method       | Num. of Parameters |  Uniform   | Fixed Adv. |
> | ------------ | :----------------: | :--------: | :--------: |
> | POMO\_AT (3) |       3.81 M       |   0.255%   |   0.295%   |
> | POMO\_AT (5) |       6.35 M       |   0.215%   |   0.257%   |
> | POMO\_AT (7) |       8.89 M       |   0.187%   |   0.242%   |
> | POMO\_AT (9) |      11.43 M       |   0.177%   |   0.231%   |
> | CNF (3)      |       3.96 M       |   0.118%   |   0.236%   |
> | CNF (5)      |       6.60 M       | **0.066%** | **0.223%** |

---

> ### Author Response · Authors · 2023-11-17
> **Response to Reviewer APh6: Part 2**
>
> **W3: The basis for selecting an attacker, and its generality.**
>
> As we discussed in the related work section (Appendix A), there exist three attackers in VRPs [1-3]. We select the attackers in our experiments based on the generality of attackers in VRPs:
>
> * The early work [2] explicitly investigates the adversarial robustness in COPs. However, their perturbation model needs to be sound and efficient, which means that, given a clean instance and its optimal solution, the optimal solution to the adversarial instance should be directly derived without running a solver. Such direct derivation requires unique properties and theorems of certain problems (e.g., the intersection theorem [6] in the Euclidean space for TSP). Hence it is non-trivial to be generalized to more complicated VRPs (e.g., CVRP). Moreover, the perturbation model in [2] is limited to attack the supervised neural solver (i.e., ConvTSP [7]), since it needs to construct the adversarial instance by maximizing the loss function so that the model prediction is maximally different from the derived optimal solution. But in the VRP domain, it is well known that the DRL models are more appealing, which do not need the optimal solution and gain even better performance, and hence attacking strong DRL-based neural solvers for VRPs is more favorable, e.g., POMO [5] used in this paper.
>
> * The work in [3] requires that the optimal solution to the adversarial instance is no worse than that to the clean instance in theory, which may limit the problem space of adversarial instances. It focuses on graph-based COPs (i.e., asymmetric TSP in [3]) and satisfies the requirement by lowering the cost of edges. Similar to [2], their method is not easy to design for VRPs with more constraints. Moreover, it resorts to the black-box adversarial attack method by training a reinforcement learning based attacker, which may lead to higher computational complexity and relatively low success rate of attacking.
>
> Therefore, for better generality, we choose [1] as the attacker in the main paper. Please note that the attacker in [1] originally attacks the model in [4] on TSP, and in this paper, we use it to attack the model in [5] on TSP and CVRP. These experiments verify its strong generality to be used for different problems and models [4, 5]. Moreover, we also evaluate the versatility of the proposed CNF against [3] as presented in Appendix B.3 and D.4.
>
> [1] Zhang, Zeyang, et al. "Learning to solve traveling salesman problem with hardness-adaptive curriculum." In AAAI 2022.
> [2] Geisler, Simon, et al. "Generalization of neural combinatorial solvers through the lens of adversarial robustness." In ICLR 2022.
> [3] Lu, Han, et al. "ROCO: A General Framework for Evaluating Robustness of Combinatorial Optimization Solvers on Graphs." In ICLR 2023.
> [4] Kool, Wouter, Herke van Hoof, and Max Welling. "Attention, Learn to Solve Routing Problems!." In ICLR 2019.
> [5] Kwon, Yeong-Dae, et al. "Pomo: Policy optimization with multiple optima for reinforcement learning." In NeurIPS 2020.
> [6] Cutler, M. "Efficient special case algorithms for the n‐line planar traveling salesman problem." *Networks* 10.3 (1980): 183-195.
> [7] Joshi, Chaitanya K., Thomas Laurent, and Xavier Bresson. "An efficient graph convolutional network technique for the travelling salesman problem." *arXiv preprint arXiv:1906.01227*(2019).

---

> ### Author Response · Authors · 2023-11-17
> **Response to Reviewer APh6: Part 3**
>
> **W4: Relationship between OOD data’s performance and the adversarial robustness.**
>
> Thanks for your suggestion. In the context of VRPs (or COPs), adversarial instances are neither anomalous nor statistical defects since they are still valid problem instances [2]. In contrast to other domains, the set of valid problems is not just a low-dimensional manifold in a high-dimensional space, and hence the manifold hypothesis [8] is not applicable to combinatorial optimization. Therefore, the concepts of generalization and robustness are somewhat correlated with each other in the context of VRPs, and it is critical for neural VRP solvers to perform well on adversarial instances while striving for better generalization performance.
>
> On the other hand, our experimental results demonstrate that raising robustness against adversarial instances by CNF favorably promotes various types of generalization of neural VRP solvers (as shown in Section 4.3 and Appendix D.3), indicating the potential existence of neural VRP solvers with high generalization and robustness concurrently. We will add the above discussion to the manuscript.
>
> [2] Geisler, Simon, et al. "Generalization of neural combinatorial solvers through the lens of adversarial robustness." In ICLR 2022.
> [8] Stutz, David, Matthias Hein, and Bernt Schiele. "Disentangling adversarial robustness and generalization." In CVRP 2019.
>
> **Q1: Effectiveness of the "global attack" instances.**
>
> Please see the results in Fig. 3 (a), where we have conducted the ablation study on TSP to verify the effectiveness of the "global attack". By comparing the result of `W/O Global Attack vs. CNF`, we have demonstrated that the global attack can greatly promote adversarial robustness, with only minor sacrifice on the standard generalization.
>
> **Q2: Will training a neural router seriously increase the training time?**
>
> The training of the neural router does not increase the time too much, since the number of parameters in the neural router is negligible, which is far less than the neural solver POMO (i.e., 0.05 million vs.1.27 million). Based on our experiments, replacing the neural router with a simple greedy heuristic (which doesn’t need any training and runs efficiently) only results in a decrease of 3 hours of the training time (~5% of the total training time). It reveals that the neural router would not bring much extra training time.

---

> ### Author Response · Authors · 2023-11-23
>
> Dear Reviewer APh6:
>
> Please kindly let us know if you still have any concerns given our response. In light of the tight deadline (today) for the author-reviewer discussion, it would be highly appreciated if you could pose them early (if any). We thank you again for your valuable comments and suggestions.

---

> > ### Comment · Reviewer_APh6 · 2023-11-23
> >
> > Thanks for the author's reply. After considering the comments of other reviewers and the content of the reply, I decided to maintain my original score.

---

> > > ### Author Response · Authors · 2023-11-23
> > >
> > > Dear Reviewer `APh6`:
> > >
> > > Thanks for your feedback. It is your right to not adjust the score.
> > > However, we think we have fixed all the issues mentioned by the reviewer. May we know the justification behind this decision, and which aspect could we improve our paper further?
> > >
> > > Thanks in advance.

---

### Official Review · Reviewer_8oTe · 2023-11-08

**Soundness:** 2 fair
**Presentation:** 3 good
**Contribution:** 3 good
**Rating:** 5
**Confidence:** 3

**Summary:**

In this paper, the authors focused on enchance the robustness and generalization of vehicle routing problems. The proposed Collaborative Neural Framework (CNF) enhances robustness by adversarially training multiple models to work together, thus improving defense against attacks and potentially increasing generalization on clean instances. This approach is supported by experimental evidence showing that CNF effectively defends against a range of attacks and also performs well on real-world benchmark instances.

**Strengths:**

1. The exploration of robustness within the context of vehicle routing problems represents a critical area of research that has received limited attention in prior studies. This paper makes a commendable contribution to the field by addressing this gap.

2. The proposed method CNF stands out for its novelty. It builds upon the established principles of min-max optimization, fundamental to adversarial training, which ostensibly enhances the robustness of the model. The efficacy of CNF is further substantiated by the empirical results presented within the study.

**Weaknesses:**

1. It seems that the adversarial attack introduced in this model has no attack budget. While acknowledging the distinctions between Vehicle Routing Problems (VRPs) and image-based tasks, it is important to note that the intrinsic discreteness of VRPs does not preclude the assignment of an attack budget, as demonstrated in adversarial settings pertinent to GNNs. The reviewer posits that evaluating the model's performance across various attack budgets is crucial for addressing the balance between clean accuracy and robust accuracy.

2. In the context of graph-based tasks, the absence of constraints on an adversary typically facilitates a significant degradation in performance, often to levels below random chance. Nevertheless, according to Table 1, the vanilla model's performance does not seem to be significantly compromised. Could the authors elucidate the factors that might be contributing to this unexpected resilience?

3. A considerable volume of literature [1-5] suggests that adversarial data augmentation can bolster generalization capabilities, particularly when the adversarial perturbations involved are small. Should the CNF enhance generalization as well, would this imply that the adversarial attack delineated within this study is potentially suboptimal in terms of its strength?

[1] Xie et al. Adversarial examples improve image recognition. CVPR 2020.

[2] Herrmann et al. Pyramid adversarial training improves vit performance. CVPR 2022.

[3] Wen et al. Adversarial cross-view disentangled graph contrastive learning. 2022

[4] Kong et al. Robust optimization as data augmentation for large-scale graphs. CVPR 2022.

[5] Cheng et al. Advaug: Robust adversarial augmentation for neural machine translation. 2020

**Questions:**

See weaknesses

---

> ### Author Response · Authors · 2023-11-17
> **Response to Reviewer 8oTe: Part 1**
>
> We sincerely appreciate the reviewer for reviewing our paper and providing constructive comments. Our detailed responses are as follows, where `W` and `Q` refer to the weakness and question, respectively.
>
> **W1: Attack budget.**
>
> We agree with the reviewer that the evaluation of the model's performance across various attack budgets is crucial for addressing the balance between clean and robust accuracy. But we would like to note the following points, which are special in the VRP domain:
>
> **(1)** Different from other domains, there is no theoretical guarantee to ensure the invariance of the optimal solution (or ground-truth label) to a clean VRP instance, given the imperceptible perturbation. A small change on even one node may induce a very different optimal solution. Therefore, we do not see the benefit of constraining the attack budget to a very small (i.e., imperceptible) range in VRP tasks. Moreover, even with the absence of imperceptible constraints on the adversary, unlike the GNN setting, we do not observe a significant degradation on clean performance as the reviewer pointed out in `W2`. It reveals that we don’t need explicit imperceptible perturbation to restrain the changes of (clean) objective values. In our experiments, we set the attack budget within a reasonable range following the applied attack method (e.g., $\alpha \in$ [1, 100] for the attacker in [1]). Our experimental results (in Section 4 and Appendix D) show that the proposed CNF is able to achieve a favorable balance, given different attack methods and their attack budgets.
>
> **(2)** In VRPs (or COPs), all generated adversarial instances are neither anomalous nor statistical defects, since they correspond to valid problem instances, regardless of how much they differ from the clean instance [2]. In this sense, the attack budget can model the severity of a potential distribution shift between training data (e.g., clean data) and test data (e.g., those similar to adversarial data). This highlights the differences to other domains (e.g., CV), where unconstrained perturbations may lead to non-realistic or invalid data. Technically, the various attack budgets can help to generate diverse adversarial instances for training. Considering the above aspects, we believe our adversarial setting, including diverse but valid problem instances, may benefit the VRP community in developing a more general and robust neural solver. With that said, this paper could also be viewed as an attempt to improve the generalization of neural VRP solvers from the perspective of adversarial robustness.
>
> [1] Zhang, Zeyang, et al. "Learning to solve traveling salesman problem with hardness-adaptive curriculum." In AAAI 2022.
> [2] Geisler, Simon, et al. "Generalization of neural combinatorial solvers through the lens of adversarial robustness." In ICLR 2022.

---

> ### Author Response · Authors · 2023-11-17
> **Response to Reviewer 8oTe: Part 2**
>
> **W2: Factors that might be contributing to the unexpected resilience (when the constraints on an adversary are absent)**
>
> Thanks for raising the question! Despite the absence of imperceptible constraints on an adversary, unlike in GNN setting, the standard generalization (i.e., clean performance) of the vanilla model `POMO_AT (1)` is not significantly degraded in VRPs (i.e., 0.144% -> 0.365% on TSP100). We would like to explain this interesting observation from two perspectives:
>
> **(1)** Different training paradigms: a popular neural VRP solver (~1 million parameters) trains the solution construction policy by deep reinforcement learning (DRL), and constructs the solution in an autoregressive way. The solution construction for each VRP instance is a complex sequential decision-making problem, formulated as a Markov Decision Process (MDP). The extra complexity comes from the problem-specific constraint handling by the masking scheme in the decoder. In contrast to supervised graph-based tasks (e.g., graph/node classification), which are often one-step prediction problems, solving a VRP instance needs to depend on iterative decisions. Such different decision structures may induce the unexpected resilience observed from the empirical results.
>
> **(2)** Desipte a VRP instance is defined over a fully-connected graph, recent studies [3, 4] show that, in each step, the decision-making by a neural solver could benefit from the local information. Specifically, the neighboring nodes (e.g., w.r.t. the Euclidean distance) of the currently selected node are more crucial to generalize the trained policy to out-of-distribution instances. From this perspective, we explain that no matter how much the adversarial instance differs from the clean one from the global view, their similar or shared local topologies in each step may be explored and leveraged by the neural solver, which, combining with the complex nature of sequential decision-making, may may contribute to the unexpected resilience.
>
> [3] Gao, Chengrui, et al. "Towards Generalizable Neural Solvers for Vehicle Routing Problems via Ensemble with Transferrable Local Policy." *arXiv preprint arXiv:2308.14104* (2023).
> [4] Generalizable Deep RL-Based TSP Solver via Approximate Invariance. Submitted to ICLR 2024.

---

> ### Author Response · Authors · 2023-11-17
> **Response to Reviewer 8oTe: Part 3**
>
> **W3: Should the CNF enhance generalization as well, would this imply that the adversarial attack delineated within this study is potentially suboptimal in terms of its strength?**
>
> Thanks for pointing out the related works [a-e], which correspond to works [1-5] mentioned by the reviewer in the original review. We have added a discussion on them in Appendix E. In addition, we also add a discussion on the selection basis of VRP attackers in Appendix E. We would like to answer this question from two perspectives:
>
> **(1)** **The source of the generalization capability enhanced by CNF:** As we discussed, in the context of VRPs (or COPs), adversarial instances are neither anomalous nor statistical defects since they correspond to valid problem instances. In contrast to other domains, the set of valid problem instances is not just a low-dimensional manifold in a high-dimensional space, and hence the manifold hypothesis in [5] does not apply to combinatorial optimization. In summary, the concepts of generalization and robustness are somewhat correlated with each other in the context of VRPs, and it is critical for neural VRP solvers to perform well on adversarial instances while striving for better generalization performance [2]. Since the proposed CNF facilitates the generation of more diverse instances during training, and reasonably exploits the overall capacity of models. Therefore, it is expected that CNF could enhance the generalization capability of neural VRP solvers.
>
> **(2)** **The strength of the adversarial attacks:** (a) If the reviewer refers to the strength of the proposed global attack, we would like to note that the empirical results of the ablation study in Figure 3 (a) (i.e., `CNF vs. W/O Global Attack`) have shown the global attack could further enhance the adversarial robustness a lot, with only slight sacrifice of the standard generalization, demonstrating its stength and effectiveness. (b) If the reviewer refers to the strength of the attackers [1, 6] used in this paper, we kindly refer the reviewer to our detailed response to [`Reviewer 4zRf Q1`](https://openreview.net/forum?id=zEOnlJaRKp&noteId=92KiLHqua1). Briefly, we select the attacker in [1] in our main experiments (due to its versatility to different VRPs) and also evaluate the generalizability of the proposed CNF against the attacker in [6], as presented in Appendix D.4. Based on our empirical results and those reported in their original papers, the attacker in [1] could degrade neural solvers more significantly than others (e.g., 0.14% -> 35.80% on TSP100). Please note that such performance degradation can be viewed as a strong attack to neural VRP solvers, since the performance is already inferior to that of naive insertion heuristics (e.g., the random insertion gains ~10% gap on TSP100, far less than 35.80%).
>
> [1] Zhang, Zeyang, et al. "Learning to solve traveling salesman problem with hardness-adaptive curriculum." In AAAI 2022.
> [2] Geisler, Simon, et al. "Generalization of neural combinatorial solvers through the lens of adversarial robustness." In ICLR 2022.
> [5] Stutz, David, Matthias Hein, and Bernt Schiele. "Disentangling adversarial robustness and generalization." In CVRP 2019.
> [6] Lu, Han, et al. "ROCO: A General Framework for Evaluating Robustness of Combinatorial Optimization Solvers on Graphs." In ICLR 2023.

---

> ### Author Response · Authors · 2023-11-23
>
> Dear Reviewer 8oTe:
>
> Please kindly let us know if you still have any concerns given our response. In light of the tight deadline (today) for the author-reviewer discussion, it would be highly appreciated if you could pose them early (if any). We thank you again for your valuable comments and suggestions.

---

### Official Review · Reviewer_ko2G · 2023-11-10

**Soundness:** 2 fair
**Presentation:** 3 good
**Contribution:** 3 good
**Rating:** 6
**Confidence:** 4

**Summary:**

This paper proposed a novel adversarial training based framework (CNF) on VRP tasks by generating global adversarial examples on a set of collaborative formed models, and distributing these samples through a well-designed attention-based neural router to perform effective joint adversarial training. The proposed method, CNF, is claimed to achieve better performance on both accuracy and robustness perspective. Experiments has been conducted by comparing CNF with various baselines on TSP and CVRP tasks. Results looks promising - CNF (3) achieves much better vanilla accuracy and robustness with large gap compared with existing baselines. Extensive ablation studies and analysis has been well presented to show the effectiveness of CNF. The whole paper contains enough experimental details and the overall writing is clear.

**Strengths:**

- The overall novelty is okay but it is interesting to see when it is applied to VRP tasks.
- Impressive experimental results - proposed CNF is better than other AT baslines by a quite large margin on TSP task.
- Detailed experimental setup and solid experimental analysis. I really love reading the Section 4.3 OOD generalization part - most of the robust training algorithm do not consider such scenario.
- The whole paper is well-written. Especially for Section 3, it is well-organized for reader to follow the exact CNF pipeline.

**Weaknesses:**

- No training efficiency was discussed while compared with other baselines. According to Algorithm 1, the training cost is largely relied on n, k and for neural router, its inference time is also heavily relied on K. We should have a column showing the exact training time for CNF and other baselines to achieve the table numbers.
- Notation is quite unclear: in Algorithm 1, n refers to the iterating variable from 1 to B and for later sections, n refers to the total number of generated samples. In Section 3.2, the captital N was introduced to refer the total number of instances. Also k refers to the attack steps while the capital K refers to the topK samples selected for neural router. However, in Section 4.2, the calligraphic K is also used to refer the topK parameter. This makes reader get confused while checking the experimental details.

**Questions:**

- (Included in Weakness part) Can you provide the total training time for both CNF and other baselines shown in Table 2?
- You include the diversity-enhanced ensemble training methods (GAL) as one of baseline methods. However, it has been proved to be not that strong compared with other recent robust ensemble training methods, such as ADP, DVERGE, TRS. Especially for DVERGE, it also claimed to have well-balance between benign accuracy and robust accuracy by crafting adversarial examples on each submodels vulnerability region and reducing adversarial transferability between submodels. It would be interesting to see how DVERGE would perform on the VRP tasks.
- I'm still quite confused about your inference setup. Given M models you have, it looks unfair to just report the best gap among all models on each instance as the actual robustness: attacker should have the prior information of each models' vulnerability to the attack instance so you should report the worst gap instead. From your side, you do not have the information about the ground-truth so you cannot always choose the best performed model against the unseen attack instances.


=============================================================

Updates:

I thank authors for conducting additional experiments and further clarifying paper notations. These new results and experimental details largely addressed my major concerns. However, I'm still quite confused about the inference reporting metric (Q.3): considering the white-box attack setting, attacker should have all information about your routing strategy and generate adv instances for the whole system instead of the best-performed model. Author should elaborate more on this. I will keep my score but raise my confidence to 4.

---

> ### Author Response · Authors · 2023-11-17
> **Response to Reviewer ko2G: Part 1**
>
> We sincerely appreciate the reviewer for spending valuable time in reviewing our paper and providing constructive comments. Our detailed responses are as follows, where we adopt `W` and `Q` to mark responses to weaknesses and questions, respectively.
>
> **W1 & Q1: Training efficiency.**
>
> Thanks for your suggestion. We show the training time of each method on TSP100 below. Please note that `POMO (1)` is the [POMO](https://github.com/yd-kwon/POMO) model (with ~1.27 million parameters) pretrained to the convergence on clean instances. All other methods are initialized from `POMO (1)` and use the same number of clean instances in the subsequent adversarial training, for the sake of fairness.
>
> | Method                    | Training Time (hrs) |
> | ------------------------- | :-----------------: |
> | POMO (1)                  |   ~11 (days) x 24   |
> | POMO\_AT (1)              |         10          |
> | POMO\_AT (3)              |         32          |
> | POMO\_HAC (3)             |         34          |
> | POMO\_DivTrain (3)        |         66          |
> | POMO_TRS (3)              |         74          |
> | CNF (3)                   |         60          |
> | CNF (3) W/O Global Attack |         42          |
> | CNF (3) W/O Router        |         57          |
>
> As mentioned in Section 5, our method does increase the computational complexity of training, due to the need for synergistically training multiple models. From the table above, we observe that our method `CNF (3)` takes longer time than the simple AT variants (i.e., POMO_AT (1), POMO_AT (3), POMO_HAC (3)). However, we note that simply further training these methods cannot significantly increase their performance. For example, we try to train POMO_AT (3) for 60 hours, with the gap still inferior to ours (i.e., `POMO_AT (3) vs. CNF (3)`: `Uniform` - 0.246% vs. 0.118%; `Fixed Adv` - 0.278% vs. 0.236%). On the other hand, our training time is less than advanced AT training methods, such as POMO_DivTrain (3) and POMO_TRS (3). The above comparison indicates that our method can achieve a better trade-off to deliver better results within reasonable training time.
>
> Inspired by the reviewer's suggestion, we also test the influence of the global attack and neural router on the training efficiency. We observe that the global attack generation seems to consume more training time than the neural router (~0.05 million parameters). we will add the full version of the effect of the number of models, the attack step, and the problem scale on the training efficiency to the final version of the manuscript.
>
> Moreover, during inference (or testing), our computational complexity only depends on the number of models (i.e., $M$), since the neural router is only trained and used during the training stage. Therefore, all methods listed above have almost the same inference time (except for those with a single model).
>
> **W2: Unclear notations.**
>
> Thanks for pointing it out. Following your comments, we have revised the notations and the corresponding figures (e.g., Fig. 2), and updated the description for better clarity. Please refer to the updated manuscript and let us know if there is still any confusion.
>
> | Notation Representation                                      |   Previous    |       Now        | Remark                                                       |
> | :----------------------------------------------------------- | :-----------: | :--------------: | :----------------------------------------------------------- |
> | Training steps in Alg. 1                                     |     $i,I$     |      $e,E$       | The $e_\mathrm{th}$ training step, $e \in [1, \dots, E]$     |
> | Iterating variable from 1 to B in Alg. 1                     |      $n$      |       $i$        | The $i_\mathrm{th}$ instance in one batch, $i\in [1,\dots,B]$ |
> | The number of customer nodes of each instance (i.e., problem scale) in Section 2.1 and Section 4 |      $n$      |       $n$        | See below                                                    |
> | Total number of instances in Section 3.2                     |      $N$      |       $N$        | $n$ and $N$ should be distinguishable, and hence we keep them the same |
> | Attack steps in Alg. 1                                       |      $k$      |       $T$        | The $t_\mathrm{th}$ attack step, $t \in [1, \dots, T]$       |
> | TopK samples selected for neural router                      |     TopK      | Top$\mathcal{K}$ | See below                                                    |
> | TopK parameter in Section 4.2                                | $\mathcal{K}$ |  $\mathcal{K}$   | The last two notations are conceptually the same, hence we use the calligraphic K for both |

---

> ### Author Response · Authors · 2023-11-17
> **Response to Reviewer ko2G: Part 2**
>
> **Q2: How about other diversity-enhanced ensemble training methods?**
>
> Thanks for your suggestion. In this paper, we focus on combinatorial optimization problems (COPs), where reinforcement learning (RL) is more appealing and effective than supervised learning (SL). The reason is that obtaining optimal solutions (i.e., ground-truth labels) for considerable instances is extremely expensive and impractical. Therefore, we choose the RL-based method - POMO in our main experiments, which is the SOTA neural construction VRP solver. On the other hand, we have read the papers [1-3] mentioned by the reviewer, which are more related to tasks in the continuous image domain. Some of them cannot be trivially adapted to VRP tasks due to the need for ground-truth labels or the dependence on the imperceptible perturbation model (which should be relaxed in VRPs as we discussed in Section 2.2).
>
> Concretely, **(1)** ADP [1]: the calculation of the ensemble diversity (Eq. (3) in [1]) requires ground-truth labels. **(2)** DVERGE [2]: it also needs input-label pairs (i.e., a target input-label pair $(x, y)$ and an independent source pair $(x_s, y_s)$) to approximate the feature distillation objective. Moreover, the distilled feature is expected to be visually similar to $x_s$ rather than $x$ but classified as the target class $y$. Such misalignment between the visual similarity and the classification result is dependent on the imperceptible perturbation model. **(3)** TRS [3]: similar to GAL [4], it proposes to use the gradient similarity loss (an improved version of [4]), and another model smoothness loss to improve ensemble robustness.
>
> Among the above methods, we find that only TRS does not need ground-truth labels and has the potential to be directly adapted to our problem setting. Following the reviewer’s suggestion, we conduct extra TSP100 experiments based on [TRS](https://github.com/AI-secure/Transferability-Reduced-Smooth-Ensemble/blob/main/train/Empirical/trainer.py#L53). However, we fail to run it on a NVIDIA-V100 GPU due to huge memory consumption, since it needs to keep the computational graph for all submodels before taking the optimization step. Therefore, we run it on an A100 GPU and show the result in the below table. Compared with other methods, we observe TRS can achieve better standard generalization but much worse adversarial robustness, and hence fail to achieve a satisfactory trade-off in our setting. Finally, we supplement a discussion about the works mentioned by the reviewer [1-4] in the `Advanced AT Techniques` paragraph of Appendix E.
>
> | Method                  | Uniform (100) | Fixed Adv. (100) |
> | ----------------------- | :-----------: | :--------------: |
> | POMO (1)                |    0.144%     |     35.803%      |
> | POMO_AT (3)             |    0.255%     |      0.295%      |
> | POMO_HAC (3)            |    0.135%     |      0.344%      |
> | POMO_DivTrain / GAL (3) |    0.255%     |      0.297%      |
> | POMO_TRS (3)            |  **0.098%**   |      0.528%      |
> | CNF (3)                 |    0.118%     |    **0.236%**    |
>
> [1] Pang, Tianyu, et al. "Improving adversarial robustness via promoting ensemble diversity." In ICML 2019.
> [2] Yang, Huanrui, et al. "DVERGE: diversifying vulnerabilities for enhanced robust generation of ensembles." In NeurIPS 2020.
> [3] Yang, Zhuolin, et al. "TRS: Transferability reduced ensemble via promoting gradient diversity and model smoothness." In NeurIPS 2021.
> [4] Kariyappa, Sanjay, and Moinuddin K. Qureshi. "Improving adversarial robustness of ensembles with diversity training." *arXiv preprint arXiv:1901.09981* (2019).
>
> **Q3: Inference setup, Why not report the worst gap instead?**
>
> During inference, we view the CNF (with $M$ submodels) as a whole system. **Each test instance will be forward passed to each submodel, and we will get $M$ solutions in total.** After that, obtaining the costs (e.g., the route length) of solutions is very efficient and straightforward (i.e., by summing the Euclidean lengths of selected edges). Since we focus on the minimization optimization problem, we choose the solution with the lowest cost (i.e., the best solution) as the final solution to the test instance, and record the corresponding best gap. We average such gaps for each test dataset and report it in our experiments. In this case, we do not require each submodel to perform well on a given (clean or adversarial) test instance, since as long as one submodel performs well is enough in our setting.

---

> ### Author Response · Authors · 2023-11-23
>
> Dear Reviewer ko2G:
>
> Please kindly let us know if you still have any concerns given our response. In light of the tight deadline (today) for the author-reviewer discussion, it would be highly appreciated if you could pose them early (if any). We thank you again for your valuable comments and suggestions.

---

> ### Author Response · Authors · 2023-11-23
> **Further reply to the reviewer's question**
>
> We really thank the reviewer for providing further feedback. Please see our response below:
>
> > However, I'm still quite confused about the inference reporting metric (Q.3): considering the white-box attack setting, the attacker should have all information about your routing strategy and generate adv instances for the whole system instead of the best-performed model. Author should elaborate more on this.
>
> Thanks for your comment. Actually, during the inference, the adversarial test dataset is generated by attacking all the models instead of the best-performed model, so as to achieve a fair evaluation (i.e., `Fixed Adv.` and `Adv.` in Table 1 mimic the black-box and white-box settings, respectively). Btw, the neural router is only used for training and is discarded (or not activated) during evaluation, since the motivation of designing the neural router is to reasonably exploit the overall model capacity during training, so as to mitigate the trade-off.
>
> Let's take the while-box (Adv.) setting as an example, given a clean instance and $M$ models in CNF, we generate the (adversarial) test instances by attacking each model independently, obtaining $M$ test instances. In other words, the adversarial test dataset includes adversarial instances generated by attacking the whole system (including the worst-performed model). We think this setting is in line with that used in ensemble-based adversarial training (such as GAL/TRS mentioned by the reviewer). We will include more details of the inference setup in the final version.
>
> May we know if our response well addresses the reviewer's concern on Q.3? We welcome further discussion if it is still not clear. Thank you!

---

### Author Response · Authors · 2023-11-17
**General Response to All Reviewers**

We sincerely appreciate all reviewers' time and efforts in reviewing our paper. Following constructive feedback from reviewers, we have updated our manuscript, which mainly includes the updated notations, additional experiments, and further discussions required by each reviewer. Specifically, we revise our paper in the following aspects.

* [Clarification] We clarify our experimental setting (e.g., inference setup; attack budget). We believe our adversarial setting, including diverse but valid problem instances, may benefit the VRP community in developing a more general and robust neural solver.
* [Notation] We give a formal definition of adversarial examples in VRPs, and update the notations in the main paper for better clarity.
* [Discussion] We add more discussions about the training efficiency, attack budget setting, selection basis of attackers, and the relationship between generalization and robustness in VRPs to Appendix E. We also additionally reviewed several related works suggested by the reviewers.
* [Extra Experiments] Following the reviewers' suggestions, we add comparative results on the training efficiency, and compare with another diversity-enhanced ensemble adversarial training method (i.e., TRS).

Please kindly see the updated manuscript and reevaluate our work. Throughout all comments, we observe that reviewers acknowledge different aspects of this paper such as the novelty (Reviewer `ko2G`, `8oTe`), writing (Reviewer `ko2G`, `4zRf`), contribution (Reviewer `ko2G`, `8oTe`), and extensive empirical experiments (All Reviewers). The questions raised by reviewers mainly stem from the specialty of the robustness study in the VRP domain, which is different from other domains (e.g., computer vision or GNN). For example, different from other domains, adversarial instances in VRPs correspond to valid problem instances, and it is critical for neural VRP solvers to perform them while aiming for better generalization performance.

We provide detailed pointwise responses to each reviewer below, and hope they can well address all reviewers' questions and concerns. We welcome any further questions or discussions if any point is unclear. Looking forward to your reply!

The authors of Paper167.

---

### Author Response · Authors · 2023-11-21
**A kind reminder for Author-Reviewer discussion**

Dear Reviewers,

We appreciate your efforts and time in reviewing our paper. We have carefully considered and responded to your initial concerns regarding our submission. As the deadline for the author-reviewer discussion period is approaching (`Nov 22`), would you mind checking our rebuttal and confirming if you have further questions? If any, let us discuss them in the OpenReview system.

Best regards,
Authors.

---

### Author Response · Authors · 2023-11-23

Dear Reviewers, AC and SAC:

We have tried our best to address all reviewers' concerns and questions. The Author-Reviewer discussion period is going to end soon. If you still have concerns, please feel free to let us know.

We hope all of you happy Thanksgiving, and may have a fruitful discussion during the *online Reviewer-AC meeting* afterwards!

Kind regards,
Authors

---

### Meta-Review · Area_Chair_TX4o · 2023-12-05

**Metareview:**

This paper studies the problem of robust neural methods for vehicle routing. Reviewers appreciate the authors' effort on the experiment. However, reviewers still have concerns on the threat model, including attack budget and the inference reporting metric.

AC has led the discussion with the reviewers. After the rebuttal phase, the reviewers are hesitant to change their opinion and no reviewer is willing to champion the paper. AC believes the paper has potential. However, the current version is marginally below the ICLR bar. AC encourages the authors to revise their paper accordingly in preparation for a future submission.

**Justification For Why Not Higher Score:**

After the rebuttal phase, the reviewers are hesitant to change their opinion and no reviewer is willing to champion the paper. The average score is clearly not above the ICLR bar.

**Justification For Why Not Lower Score:**

N/A

---

### Decision · Program_Chairs · 2024-01-16

Reject